# Co-emission of volcanic sulfur and halogens amplifies volcanic effective radiative forcing

John Staunton-Sykes[*1], Thomas J. Aubry[2,4], Youngsub M. Shin[1], James Weber[1], Lauren R. Marshall[1], Nathan Luke Abraham[1,3], Alex Archibald[1,3], Anja Schmidt[1,2]

[1]*Centre for Atmospheric Science, Department of Chemistry, University of Cambridge, Cambridge, UK*

[2]*Department of Geography, University of Cambridge, Cambridge, UK*

[3]*National Centre for Atmospheric Science, UK*

[4]*Sidney Sussex College, Cambridge, UK*

*Correspondence to*: J. Staunton-Sykes (email:jjas3@cam.ac.uk)

**Abstract**

The evolution of volcanic sulfur and the resulting radiative forcing following explosive volcanic eruptions is well understood. Petrological evidence suggests that significant amounts of halogens may be co-emitted alongside sulfur in some explosive volcanic eruptions, and satellite evidence indicates that detectable amounts of these halogens may reach the stratosphere. In this study, we utilise an aerosol-chemistry-climate model to simulate stratospheric volcanic eruption emission scenarios of two sizes, both with and without co-emission of volcanic halogens, in order to understand how co-emitted halogens may alter the life cycle of volcanic sulfur, stratospheric chemistry and the resulting radiative forcing. We simulate a large (10 Tg of $SO_2$) and very large (56 Tg of $SO_2$) sulfur-only eruption scenario and a corresponding large (10 Tg $SO_2$, 1.5 Tg HCl, 0.0086 Tg HBr) and very large (56 Tg $SO_2$, 15 Tg HCl, 0.086 Tg HBr) co-emission eruption scenario. The eruption scenarios simulated in this work are hypothetical, but they are comparable to Volcanic Explosivity Index (VEI) 6 (e.g. 1991 Mt. Pinatubo) and VEI 7 (e.g. 1257 Mt. Samalas) eruptions, representing 1 in 50-100 year and 1 in 500-1000 year events respectively, with plausible amounts of co-emitted halogens based on satellite observations and volcanic plume modelling.

We show that co-emission of volcanic halogens and sulfur into the stratosphere increases the volcanic effective radiative forcing (*ERF*) by 24% and 30% in large and very large co-emission scenarios compared to sulfur-only emission. This is caused by an increase in both the forcing from volcanic aerosol-radiation interactions (*ERF_{ari}*) and composition of the stratosphere (*ERF_{clear,clean}*). Volcanic halogens catalyse the destruction of stratospheric ozone, which results in significant stratospheric cooling, offsetting the aerosol heating simulated in sulfur-only scenarios and resulting in net stratospheric cooling. The ozone induced stratospheric cooling prevents aerosol self-lofting and keeps the volcanic aerosol lower in the stratosphere with a shorter lifetime. This results in reduced growth by condensation and coagulation, and smaller peak global-mean effective radius compared to sulfur-only simulations. The smaller effective radius found in both co-emission scenarios is closer to the peak scattering efficiency radius of sulfate aerosol, and thus, co-emission of halogens results in larger peak global-mean *ERF_{ari}* (6% and 8%). Co-emission of volcanic halogens results in significant stratospheric ozone, methane and water

vapour reductions, resulting in significant increases in peak global-mean $ERF_{clear,clean}$ (>100%), predominantly due to ozone loss. The dramatic global-mean ozone depletion simulated in large (22%) and very large (57%) co-emission scenarios would result in very high levels of UV exposure on the Earth's surface, with important implications for society and the biosphere.

This work shows for the first time that co-emission of plausible amounts of volcanic halogens can amplify the volcanic $ERF$ in simulations of explosive eruptions. It highlights the need to include volcanic halogen emissions when simulating the climate impacts of past or future eruptions, and the necessity to maintain space-borne observations of stratospheric compounds to better constrain the stratospheric injection estimates of volcanic eruptions.

## 1 Introduction

Sulfur gases emitted into the atmosphere by volcanic eruptions have a strong direct climate impact through the formation of sulfuric acid aerosol, which reflect incoming sunlight and cool the Earth's surface (Robock, 2000). Volcanic aerosols also have the potential to alter the chemistry of the stratosphere, including ozone with significant impacts on both longwave and shortwave radiative fluxes. Ozone is impacted dynamically by stratospheric circulation changes induced by aerosol heating, and chemically by changes to ozone catalytic loss cycles. Aerosol heating in the tropics increases the vertical ascent transporting ozone to higher altitudes and latitudes, resulting in an ozone decrease in the tropics and an increase at high latitudes (Kinne et al., 1992). The addition of large amounts of volcanic aerosols increases the surface area of the stratosphere on which heterogeneous reactions can take place (Solomon, 1999). Heterogeneous reactions in the stratosphere drive changes in the partitioning of $NO_x$, $ClO_x$, $BrO_x$ and $HO_x$ species between reservoir and active forms. Unlike polar stratospheric clouds (PSCs), which only occur in the extremely cold temperatures inside the winter polar vortex, volcanic aerosols provide surfaces for heterogeneous reactions at all latitudes and at all times of the year. $N_2O_5$ reacts with water vapour on the surfaces of these volcanic aerosols to form $HNO_3$. This effectively sequesters reactive $NO_x$ species into a long-lived reservoir and limits the availability of $NO_x$ radicals which take part in catalytic ozone loss reactions, reducing the chemical destruction of ozone (Crutzen, 1970). In contrast, these reactions liberate reactive $ClO_x$ and $BrO_x$ species from their long-lived reservoirs, increasing the chemical destruction of ozone (Aquila et al., 2013; Solomon, 1999; Solomon et al., 1996). The net chemical impact of volcanic sulfate aerosol loading on stratospheric ozone is dependent on the stratospheric halogen loading. A large volcanic eruption in low-halogen atmospheric conditions, such as a preindustrial or future atmosphere, is expected to result in a net stratospheric ozone increase (Langematz, 2018), however, when the halogen loading of the stratosphere is high, an eruption will lead to a net stratospheric ozone decrease (e.g. Tie & Brasseur, 1995). High-halogen loading may arise from anthropogenic or natural emissions.

Petrological data suggest that volcanic eruptions in some geological settings may also release substantial amounts of halogen gases into the atmosphere (Krüger et al., 2015; Kutterolf et al., 2013, 2015). Petrological analysis of the 1257 Mt. Samalas eruption suggests as much as 227 Tg of hydrogen chloride (HCl) and 1.3 Tg of hydrogen bromide (HBr) could have been emitted into the atmosphere alongside 158 Tg of sulfur dioxide ($SO_2$) (Vidal et al., 2016). The portion of the halogens erupted at the vent that reach the stratosphere (hereafter halogen injection efficiency) is not well constrained and has been the subject of debate in the community for decades. Halogens are

soluble (especially HCl) and may be scavenged by water, ice hydrometeors and ash in the volcanic plume (Halmer et al., 2002). Despite efficient scavenging, direct stratospheric injection of volcanic halogens is predicted theoretically, and sophisticated plume models suggest that between 10% and 20% of the HCl emitted at the vent of large explosive eruptions could reach the stratosphere (Textor et al., 2003).


Aircraft measurements following the 2000 Mt. Hekla eruption in Iceland showed that 75% of the HCl emitted at the vent entered the lower stratosphere and was still present 35 hours after the eruption, suggesting that little scrubbing took place in the tropospheric eruption column (Hunton et al., 2005; Rose et al., 2006). Read et al. (2009) used retrievals from the Microwave Limb Sounder (MLS) to show that $SO_2$ and HCl was injected directly

into the lower stratosphere during the 2004 Manam, 2007 Anatahan, 2008 Soufriere Hills, 2008 Okmok, 2008 Kasatochi, 2009 Redoubt, and 2009 Sarychev eruptions. Using retrievals from MLS, Prata et al. (2007) reported HCl at ~20 km in the volcanic plume of 2006 Soufriere Hills eruption plume, with stratospheric HCl:$SO_2$ gas ratios <0.1. Carn et al. (2016) reported MLS stratospheric HCl:$SO_2$ gas ratios of 0.01–0.03 (relative mixing ratios) for 14 small eruptions in the period between 2005 to 2014. Limitations with the field of view and spatial sampling

of MLS mean these observed ratios are likely an underestimate (Carn et al., 2016).

Petrological analysis in Bacon et al. (1992) suggested that the considerably larger, Volcanic Explosivity Index (VEI) 7, 7.6 kya eruption of Mt. Mazama degassed ~100 Tg of Cl, and the ice core record of the same eruption suggested 8.1 Tg Cl and  57.5 Tg $SO_2$ was injected into the stratosphere with a halogen injection efficiency of

8.1% and a stratospheric HCl:$SO_2$ molar ratio of ~0.3 (Zdanowicz et al., 1999). The two largest eruptions in the satellite era, 1982 El Chichón and 1991 Mt. Pinatubo, highlight the variability in stratospheric halogen injection following explosive volcanic eruptions. Both eruptions released relatively small amounts of halogens, 1.8 Tg (Varekamp et al., 1984) and 4.5 Tg of chlorine respectively, with HCl:$SO_2$ molar ratios of ~0.4 (Mankin et al., 1992). Spectroscopic measurements of the El Chichón stratospheric eruption plume indicated an HCl increase of

40% compared to measurements taken prior to the eruption, with a stratospheric injection of >0.04 Tg of HCl and a halogen injection efficiency of at least 2.5% (Mankin and Coffey, 1984; Woods et al., 1985). Woods et al. (1985) measured NaCl salt particles in the lower stratospheric eruption cloud of El Chichón derived from the chlorine-rich magma. They hypothesised that the rapid ascent of large Plinian eruption phases led to the formation of ice-bearing crystals and salt particles, which would lower the halogen scrubbing efficiency and preserve the halogens

for stratospheric release. In the stratosphere, these salt particles may react with volcanic sulfuric acid leading to the formation of secondary HCl. In contrast, despite emitting more Cl into the atmosphere than El Chichón, observations following the 1991 Mt. Pinatubo eruption showed minimal stratospheric halogen injection, due to the fact that halogens were more efficiently scavenged in the eruption cloud (Wallace and Livingston, 1992). The Pinatubo eruption occurred at the same time and in the same location as a typhoon in the Philippines, and it is

thought these very wet tropospheric conditions led to the effective wash out of halogens (Gerlach et al., 1996; McCormick et al., 1995; Self S et al., 1996).

Overall, current datasets show that the stratospheric injection of volcanic halogens is highly variable and depends on both the total mass of halogens released at the vent and the degree of scavenging, which is determined by the

geochemistry of the volcano and the prevailing atmospheric conditions during the eruption, particularly the

humidity. It is clear, however, that volcanic halogens are injected into the stratosphere after some volcanic eruptions, but there is limited research into how these volcanic halogens may alter the volcanic aerosol microphysics, stratospheric chemistry, and volcanic forcing.

Lurton et al. (2018) simulated the 2009 Sarychev Peak eruption (0.9 Tg of $SO_2$) in CESM1(WACCM) (Community Earth-System Model, Whole Atmosphere Community Climate Model) and showed how inclusion of co-emitted halogens (27 Gg of HCl) resulted in a lengthening of the $SO_2$ lifetime, due to the further depletion of OH, and a corresponding delay in the formation of aerosols, giving better agreement between modelled and observed $SO_2$ burden and showing how co-emitted halogens could impact volcanic sulfur processing.


Tie and Brasseur (1995) utilised model calculations to show how background atmospheric chlorine loadings altered the ozone response to volcanic sulfur injections. In conditions typical of the pre-1980 period, the ozone column abundance was shown to increase after a large volcanic eruption. The increase in column abundance was the result of suppression of the $NO_x$ catalysed ozone loss cycle, which was driven by the sequestration of reactive

nitrogen to its reservoir species via heterogeneous reactions on the surface of volcanic aerosol. The ozone response was shown to be independent of the magnitude of the eruption, as the heterogeneous conversion of active nitrogen to its reservoir was saturated. However, after 1980, higher background chlorine levels resulting from the anthropogenic emissions of chlorofluorocarbons, meant that the ozone response became negative in winter at mid and high latitudes. The suppression of $NO_x$ catalysed ozone loss was counterbalanced by an increase in the $ClO_x$

catalysed ozone loss, resulting in a transition in the column ozone response. Unlike in pre-industrial conditions, the ozone response was dependent on the eruption size as the heterogeneous conversion of chlorine species from reservoir to reactive was not saturated. Since then, a number of studies have investigated the impact of volcanic halogens on stratospheric ozone. Cadoux et al. (2015) petrologically determined chlorine and bromine degassing budgets for the Bronze Age (~1600 BCE) Santorini eruption and, using a halogen injection efficiency of 2%, input

36 Tg S, 13.5 Tg Cl and 0.02 Tg Br uniformly between the tropopause and 35 km in a pre-industrial background state within a 2D chemical transport model (CTM). They simulated ozone depletion lasting a decade with a peak global-mean of 20-90% over the northern hemisphere. The molar ratio of HCl and $SO_2$ injected into the stratosphere ($HCl:SO_2$) in this study was 0.64, considerably larger than observations from MLS (<0.1) and ice core records of Mt. Manzana (<0.3). Klobas et al. (2017) also used a 2D CTM to study the impact that co-emission

of volcanic halogens has on column ozone in contemporary and future background states. They simulated hypothetical Pinatubo sized eruptions with a $HCl:SO_2$ of ~0.14 and reported global ozone depletion lasting ~2-3 years with a peak of 20%. These CTM studies used prescribed wind fields and, as a result, do not include the important interactive feedbacks of radiation and dynamics which alter the transport of tracers and thus the composition of the atmosphere. Ming et al. (2020) simulated explosive tropical eruptions in a chemistry-climate

model which consisted of the UK Met Office Unified Model (UM) together with the United Kingdom Chemistry and Aerosol (UKCA) scheme, including the interactive stratospheric aerosol model GLOMAP-mode (Mann et al., 2010). They simulated 6 sets of experiments: low $SO_2$ (10 Tg) and high $SO_2$ (100 Tg) eruptions paired with no HCl, low HCl (0.02 Tg) and high HCl (2 Tg), and reported significant ozone depletion over both poles for at least four years in the high $SO_2$ and high HCl experiment. Brenna et al. (2019) used CESM1(WACCM) with

prescribed volcanic aerosols and sea surface temperatures (SSTs) to simulate an average eruption of a Central

American Volcanic Arc volcano in a pre-industrial background state, with a 10% halogen injection efficiency (2.5 Tg Cl, 9.5 Gg Br). They found ozone depletion of up to 20% globally for 10 years, with ozone hole conditions over the tropics and Antarctica. Consequently, UV radiation increases of >80% were simulated in the tropics, averaging to >40% for 2 years.


However, these studies did not investigate how volcanic halogens may interact with the sulfur aerosol life cycle and modulate volcanic forcing. Brenna et al. (2020) used Community Earth System Model 2 (WACCM6) to investigate the coupling and feedback between volcanic aerosol, chemistry, radiation and climate pre-industrial background state. They investigate the combined effect of the sulfur (523 Tg S) and halogens (120 Tg Cl, 0.2 Tg
Br) emissions of the Los Chocoyos super-eruption, assuming a 10% halogen injection efficiency resulting in a stratospheric $HCl:SO_2$ molar ratio ~0.4, on volcanic gases, ozone and surface UV. Compared to simulations with sulfur-only injections, they simulate a lower peak sulfate burden attributed to the delay in $SO_2$ oxidation but with the same total sulfur lifetime and aerosol effective radius. Thus, the co-emission of halogens results in a smaller radiative forcing; 20% lower compared to sulfur-only. Wade et al. (2020) compared HadGEM3-ES (Earth System
configuration of the Hadley Centre Global Environment Model version 3) simulations of the 1257 Mt. Samalas eruption, utilising the halogen degassing estimates from Vidal et al. (2016) and stratospheric halogen injection efficiencies of 20% and 1%, with the available surface temperature proxies. Their results suggest it is unlikely that 20% of degassed halogens reached the stratosphere, but smaller fractions gave good agreement with multi-proxy surface temperature records.


The aim of this study is to simulate hypothetical large and very-large sized eruptions, both with and without halogens, in a coupled chemistry-aerosol model in order to investigate how the co-emission of volcanic sulfur and halogens alters the evolution of volcanic aerosol, ozone, stratospheric composition, and the consequential radiative forcing and UV flux.

**2 Data and Methods**

**2.1 Model Description**

This study uses UKESM-AMIP, the atmosphere-only configuration of the UK Earth System Model UKESM1.0 (Sellar et al., 2019) including coupled aerosol-chemistry-climate components consisting of the United Kingdom Chemistry and Aerosol (UKCA) module together with the UK Met Office Unified Model (UM). The UKCA
module is run at UM version 11.2 with the combined stratosphere and troposphere chemistry (StratTrop) scheme (Archibald et al., 2020). The model is free-running in the atmosphere, forced by sea ice and sea surface temperature surface boundary conditions, similar to the set up used in the UK Earth system model (UKESM1) Atmospheric Model Intercomparison Project (AMIP) simulations submitted to the Coupled Model Intercomparison Project Phase 6 (CMIP6) (Sellar et al., 2019, 2020). The resolution was 1.875° longitude by 1.25°
latitude with 85 vertical levels extending from the surface to 85 km. The dynamics of the stratosphere have previously been shown to be well represented in this model, and it has an internally generated Quasi-Biennial Oscillation QBO (Osprey et al., 2013) The model includes the fully interactive stratospheric GLOMAP-mode aerosol scheme which simulates microphysical processes including the formation, growth, transport and loss of

aerosol (Dhomse et al., 2014). GLOMAP-mode also calculates aerosol optical properties online which are used
to calculate direct and indirect radiative effects (Mulcahy et al., 2020).

In UKCA, stratospheric ozone concentrations are determined by sets of photochemical reactions as well as ozone
destroying catalytic cycles involving chlorine, bromine, nitrogen, and hydrogen radical species (Archibald et al.,
2020). Photolysis reactions in UKCA utilise rates calculated from a combination of the FAST-JX scheme and
look-up tables (Telford et al., 2013). Ozone depleting radical species are produced by the photolysis of halogen
containing compounds reacting on the surface of stratospheric aerosols, including hydrochloric acid (HCl),
chlorine nitrate (ClONO$_2$), hydrogen bromide (HBr), and bromine nitrate (BrONO$_2$). Heterogeneous reactions in
the presence of polar stratospheric clouds (PSCs) in the polar lower stratosphere or in the presence of sulfate
aerosol following explosive volcanic eruptions are also important for stratospheric ozone concentrations. Eight
additional heterogeneous reactions involving chlorine and bromine species were added as described in Ming et al.
(2020), with the main change being the explicit treatment of the reactions of four additional chemical species: Cl$_2$,
Br$_2$, ClNO$_2$, and BrNO$_2$ which are photolysed to produce Cl and Br radicals.

Volcanic effective radiative forcings (hereafter *ERF)* are calculated as differences (*Δ*) in the net top of atmosphere
(TOA) radiative fluxes (*F*) between perturbed and control climatologies as follows:

$$ERF = \Delta F \qquad Eq.\,1$$

Volcanic *ERF* is decomposed as described in (Schmidt et al., 2018) and (Ghan, 2013), as follows:

$$ERF = \Delta(F - F_{clean}) + \Delta\left(F_{clean} - F_{clear,clean}\right) + \Delta F_{clear,clean} \qquad Eq.\,2$$

$$= ERF_{ari} + ERF_{aci} + ERF_{clear,clean} \qquad Eq.\,3$$

This decomposition is enabled by implementing extra calls to the radiation scheme as recommended by Ghan
(2013) to obtain $F_{clean}$ and $F_{clear,clean}$. Where $F_{clean}$ denotes a radiation flux diagnostic calculation without aerosol-
radiation interactions but including aerosol-cloud interactions through microphysics. $F_{clear,clean}$ denotes a radiation
flux diagnostic calculation that ignores both aerosol and cloud-radiation interactions. Thus, $F$ - $F_{clean}$, determines
the impact of all aerosols and *Δ(F - $F_{clean}$)* is an estimate of the forcing from volcanic aerosol-radiation interactions
(*$ERF_{ari}$*). The second term *Δ($F_{clean}$ - $F_{clean,clear}$)* represents the difference in the clean-sky cloud radiative forcing,
and is an estimate of the aerosol-cloud interactions (*$ERF_{aci}$*) due to volcanic emissions. The third term, *$ERF_{clear,clean}$*
accounts for changes not directly due to aerosol or cloud interactions, largely the result of changes in surface
albedo and atmospheric composition.

**2.2 Experimental Design**

We utilise atmosphere-only, time-slice experiments whereby the SST, sea ice fraction and depth, surface
emissions and lower boundary conditions are prescribed using climatologies calculated using data from the fully
coupled UKESM1.0 historical runs produced for CMIP6 (Eyring et al., 2016) and averaged over the years 1990
to 2000.  By averaging over the decade the atmosphere-only simulations are forced with boundary conditions
typical of the recent historical period but not a specific date within that decade. The fully coupled transient

simulations had internally generated El Nino and La Nina cycles, however, averaging the SSTs over the 1990 to 2000 period resulted in a permanent neutral signal in the SST pattern (see Figure S1). The 1990s, and thus these timeslices, were characterised by high background halogen levels due to anthropogenic emissions of CFCs throughout the preceding decade. The impacts of very short-lived Bromine species are accounted for by adding a fixed contribution of 5 pptv into the $CH_3Br$ surface concentration.

A control simulation was initialised from the January 1995 initialisation file taken from the UKESM1.0 historical scenario which was run as part of CMIP6 (Eyring et al., 2016). The model was allowed to spin up for 15 years and a control simulation was run for a further 20 years. The effect of explosive volcanic eruptions was investigated by running a series of 10-year volcanic perturbation simulations spun off from 6 different years in the control run to represent the variability in QBO states. Changes are plotted as the difference between the average of the 6 ensembles and a climatology derived from the 20-year control run, cumulative forcings are calculated as the time-integrated forcing across the Earth's surface and represent the total energy loss (J) as a result of the volcanic eruption.

The volcanic emissions are prescribed by direct injection of $SO_2$, HCl and HBr into the stratosphere with a Gaussian vertical distribution centred on 21 km and a width of 2.1 km (10% of the height), lasting for 24 hours on July 1st. An injection altitude of 21 km was chosen as, allowing for lofting, this results in a volcanic plume altitude consistent with recent historical eruptions from the satellite era (Guo et al., 2004). The gases were injected in the tropics ($5^o$S latitude and $0^o$ longitude) to represent a typical tropical explosive eruption (Newhall et al., 2018).

Since historical stratospheric volcanic $SO_2$ fluxes are variable and the volcanic flux of HCl and HBr into the stratosphere remains uncertain, we developed a simulation matrix that spans a range of possible explosive volcanic emissions. The four sets of experiments have one large $SO_2$ (10 Tg), and one very-large $SO_2$ (56 Tg) emission scenario both with (HAL10 and HAL56) and without halogens (SULF10 and SULF56), as shown in Table 1. These eruption sizes (10 and 56 Tg $SO_2$) are hypothetical, but they are comparable to a VEI 6 (e.g. 1991 Mt. Pinatubo) and VEI 7 (e.g. 1257 Mt. Samalas) eruption, representing 1 in 50-100 year and 1 in 500-1000 year events respectively (Newhall et al., 2018). VEI is used here to provide context of the recurrence rates but is not used as an index representative of climate impact. HAL56 utilises the 1257 Mt. Samalas HCl and HBr emission estimates from Vidal et al. (2016) and assumes a conservative ~5% stratospheric halogen injection efficiency, less than the 10-20% predicted by plume modelling in Textor et al. (2013) and closer to the observed efficiency following El Chichón (>2.5%) and in the ice core record of Mt. Mazama (8%), as well as the fraction supported by Wade et al. (2020). HAL10 has a $SO_2$ injection similar to that found to reproduce the spatial and temporal evolution of stratospheric aerosol optical depth (SAOD) following 1991 Pinatubo (Mills et al., 2016) and a 10 times smaller HCl and HBr flux than HAL56. This results in a HCl:$SO_2$ ratio of ~0.26 and ~0.47 in HAL10 and HAL56 respectively, similar to the estimated stratospheric injection ratio for Mt. Mazama (0.3) (Zdanowicz et al., 1999) and the ratios used in Ming et al. (2020) and Brenna et al. (2020) but smaller than the ratio used in Cadoux et al. (2015).

| Scenario | SO$_2$ (Tg) | HCl (Tg) | HBr (Tg) | HCl:SO$_2$ |
|----------|-------------|----------|----------|------------|
| SULF56 | 56 | - | - | - |
| HAL56 | 56 | 15 | 0.086 | 0.47 |
| SULF10 | 10 | - | - | - |
| HAL10 | 10 | 1.5 | 0.0086 | 0.26 |

**Table 1** Showing the eruption masses of SO$_2$, HCl and HBr in Tg for the four sets of experiments.

**3 Results**

**3.1 Sulfur Microphysics and ERF$_{ari}$**

Atmospheric burdens of volcanic sulfur species are summarized in Figure 1. As shown by Lurton et al. (2018), volcanic halogens deplete the hydroxyl radical (OH) via equation 4

$$HCl + OH \rightarrow Cl + H_2O \qquad Eq. 4$$

which limits the availability of OH for SO$_2$ oxidation, leading to slower destruction of volcanic SO$_2$ and an increase in SO$_2$ e-folding time of 21% and 40% in HAL10 and HAL56 compared to SULF10 and SULF56 respectively. As the rate of formation of sulfuric acid is decreased, we simulate a corresponding delay in the

formation of sulfate aerosol and a reduction in the peak sulfate aerosol burden by 8% in both HAL10 and HAL56.

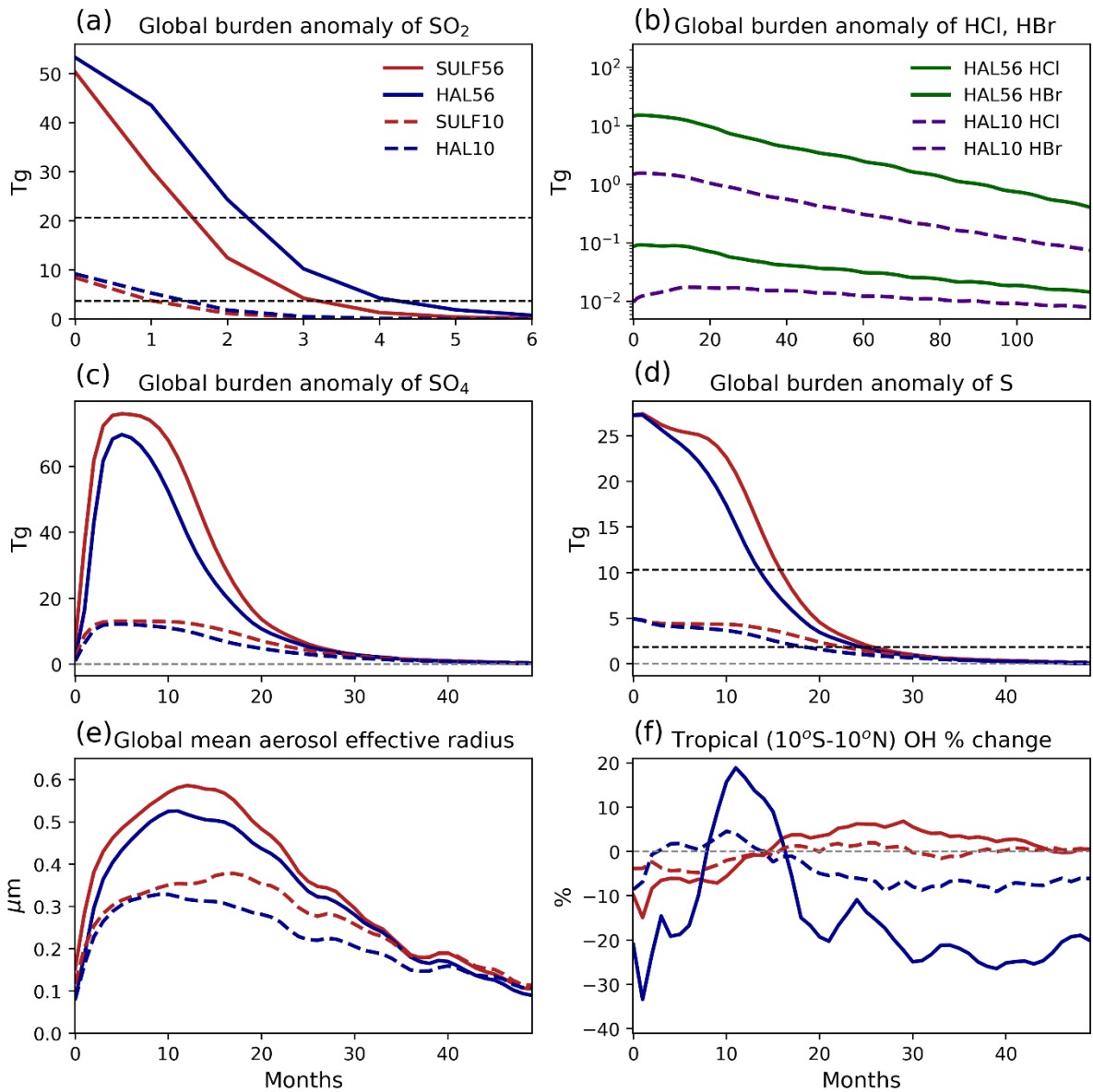

**Figure 1** Evolution of sulfur, halogens, aerosol effective radius and OH for the SULF10, HAL10, SULF56 and HAL56 simulations relative to the control climatology. **(a)** Global $SO_2$ burden anomalies. **(b)** Global HCl and HBr burden anomalies on log scale. **(c)** Global sulfate aerosol burden anomalies. **(d)** Global total sulfur burden anomalies. (e) Global-mean aerosol effective radius, weighted by aerosol surface area density. (**f**) Tropical ($10^oN$-$10^oS$) stratospheric OH change (%). Dashed horizontal lines in (a) (b) and (d) represent the mass remaining after one e-folding time. Note the different axis scales.

Despite the slower rate of $SO_2$ oxidation, the co-emission of halogens reduces the e-folding lifetime of the sulfur burden to 17.3 and 11.7 months in HAL10 and HAL56, compared with 21.2 and 13.6 months in SULF10 and SULF56, a decrease of 18% and 14% respectively. This indicates that co-emission of halogens alters the rate at which sulfur is removed from the atmosphere. Significant differences in stratospheric temperature change are simulated between the sulfur-only and halogen simulations. In sulfur-only simulations, strong positive temperature anomalies (~3 K) due to sulfate aerosol absorption of infra-red radiation are simulated across the

tropical stratosphere (Figure 2). This aerosol heating increases the vertical ascent (Figure S2) and lofts volcanic aerosol to altitudes higher than the initial injection height in the model. By contrast, co-emission of volcanic halogens results in significant stratospheric ozone depletion of 22-57% (see section 3.2) and, in turn, this results in large negative temperature anomalies (~ -3 K) over most of the lower and middle stratosphere (Figure 2). Ozone

generates heat in the stratosphere by absorbing both incoming shortwave (SW) radiation from the Sun and by absorbing upwelling longwave (LW) radiation from the troposphere. Thus, decreasing stratospheric ozone results in stratospheric cooling, offsetting the volcanic aerosol heating and resulting in net stratospheric cooling. This stratospheric cooling decreases the vertical ascent in the tropics (Figure S2) and prevents volcanic sulfate aerosol being self-lofted in HAL10 and HAL56. The volcanic sulfate aerosol thus remains at significantly lower altitudes

in HAL10 and HAL56 (~21-22 km) compared with SULF10 and SULF56 (~24-25 km) (Figure 2e,f). Lower altitude aerosol remains in a faster region of the Brewer-Dobson Circulation (Figure S3) which results in faster transport to high-latitudes and removal from the stratosphere (Figure 1d).

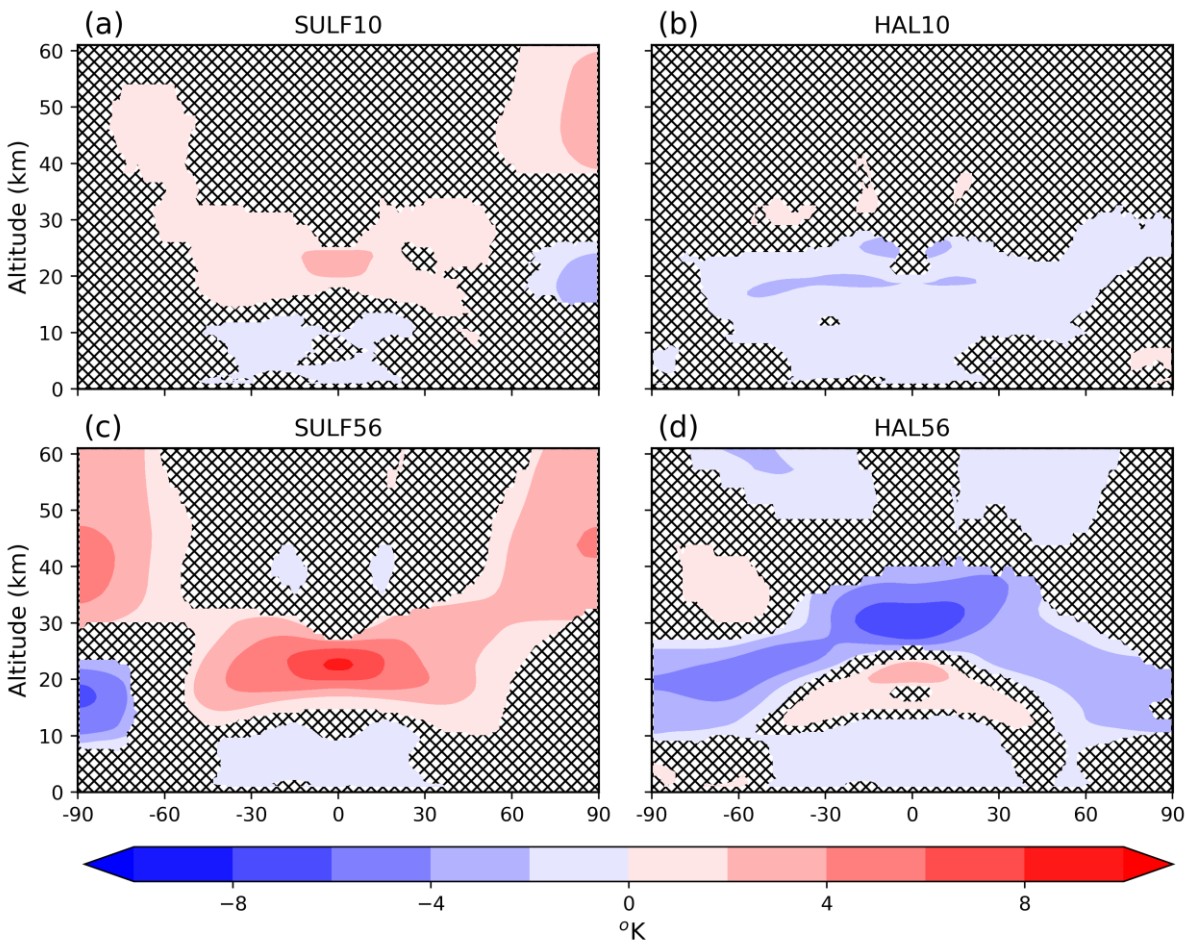

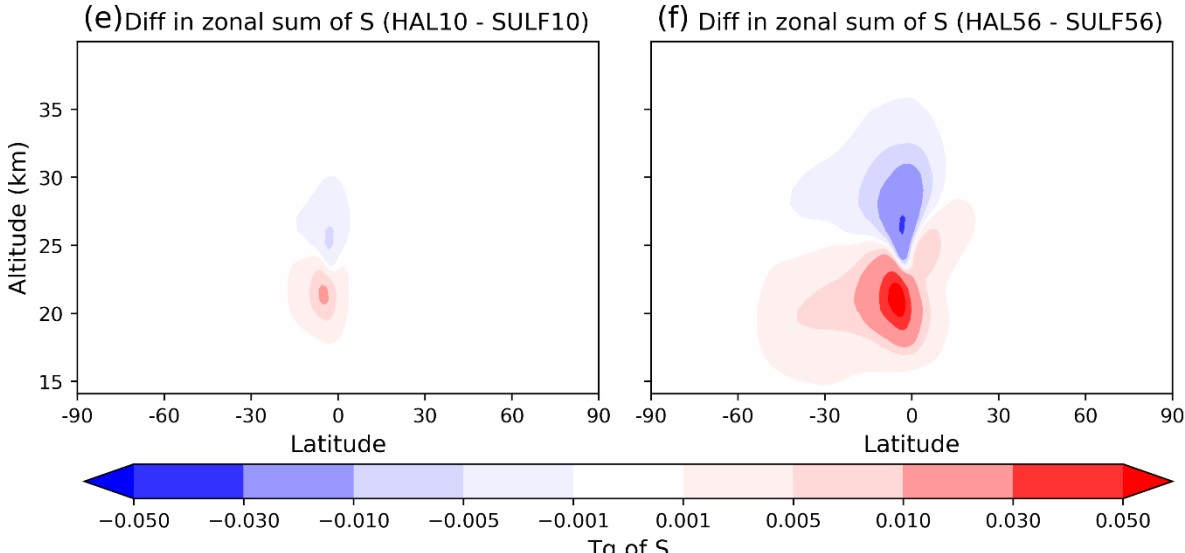


**Figure 2** Zonal-mean temperature anomaly (K) averaged over the first post-eruption year relative to the control climatology **(a)** SULF10, **(b)** HAL10, **(c)** SULF56, **(d)** HAL56. Differences that are not significant at the 95% confidence interval according to a Mann–Whitney U test are indicated with stipples. Difference in zonal sum of total sulfur averaged over the first year post-eruption **(e)** HAL10 – SULF10, **(f)** HAL56 – SULF56.


The maximum global-mean aerosol effective radii ($R_{eff}$) is 0.38 µm and 0.59 µm in SULF10 and SULF56, respectively. The maximum global-mean $R_{eff}$ simulated in SULF10 is similar to that derived from measurements following 1991 Pinatubo, with an estimate of 0.4 - 0.5 µm from balloon borne measurements (Deshler et al., 1997) and 0.45 µm obtained from GLOSSAC satellite observations (GloSSAC, version 1.1; Thomason et al., 2018).

The shorter lifetime of sulfur in the atmosphere following HAL10 and HAL56 eruptions results in reduced aerosol growth and smaller aerosol $R_{eff}$. The peak global-mean $R_{eff}$ is ~15% and ~10% smaller in HAL10 and HAL56 compared to their equivalent SULF simulations (Figure 1e). This aerosol growth stunting effect is a direct result of the shorter sulfur lifetime, rapid spreading and removal of aerosol. Volcanic sulfate aerosols grow through microphysical processes of condensation and coagulation (Kremser et al., 2016). The faster removal of sulfate

aerosol in HAL10 and HAL56 reduces the growth via condensation and coagulation and results in smaller peak global-mean aerosol $R_{eff}$. This theory is supported by Figure 3 which shows a scatter plot of the 3-year global-mean aerosol effective radius as a function of the global sulfur burden e-folding lifetime for each individual ensemble member, with a significant correlation within both 10 Tg ($r$=0.88) and 56 Tg ($r$=0.95) eruption ensembles. The positive correlation between these two variables holds only for each eruption size scenario. To a

first order, the aerosol $R_{eff}$ is determined by the magnitude of the volcanic sulfur injection. The larger SO$_2$ injection in HAL56 and SULF56 ensemble simulations leads to larger-sized sulfate aerosols, faster sedimentation and shorter removal time compared to HAL10 and SULF10 ensemble simulations. However, when we fix the mass of sulfur injected and compare sulfur-only and co-emission scenarios, we find that transport has a second order effect. The faster removal of sulfate aerosol in HAL10 and HAL56 ensemble simulations leads to smaller-sized

aerosol due to reduced opportunity for aerosol growth compared with SULF10 and SULF56 respectively.

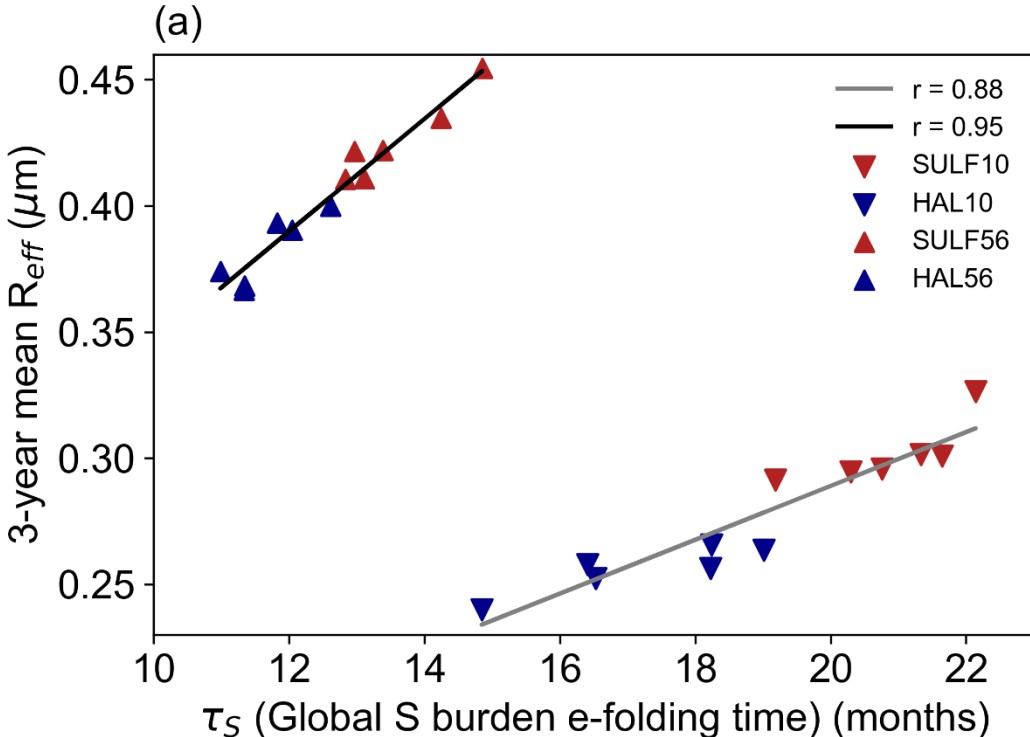

**Figure 3** Global-mean aerosol effective radius over the first 3 post-eruption years as a function of the global total sulfur e-folding time. Both plots have regression lines fitted with correlation coefficient (*r*) showing strong positive correlation.

The radiative impact of sulfate aerosols depends on the particle size (Timmreck et al., 2010). Using Mie scattering theory, Lacis (2015) found that the scattering cross section per unit mass is largest for sulfate aerosol with effective radius of ~0.20 μm. The smaller aerosol $R_{eff}$ in HAL10 and HAL56, compared to SULF10 and SULF56, is closer to 0.20 μm and results in more efficient scattering of SW radiation per unit mass (Timmreck et al., 2010). Therefore, we simulate 11% and 22% higher peak global-mean SAOD anomalies at 550 nm in HAL10 and HAL56 than their equivalent SULF simulations (Figure 4), despite having a 14% and 9% smaller peak aerosol burden. Correspondingly, we simulate an 8% and 6% increase in the peak global-mean $ERF_{ari}$ in HAL10 and HAL56 compared to SULF10 and SULF56 (Figure 4), driven by a 14% and 11% increase in peak global-mean SW forcing (Figure S4). The SAOD and $ERF_{ari}$ anomalies are a balance between the offsetting effects of smaller aerosol and shorter lifetime which result in a net-zero impact on cumulative $ERF_{ari}$ despite a significant increase in the peak global-mean $ERF_{ari}$ (Figure S5).

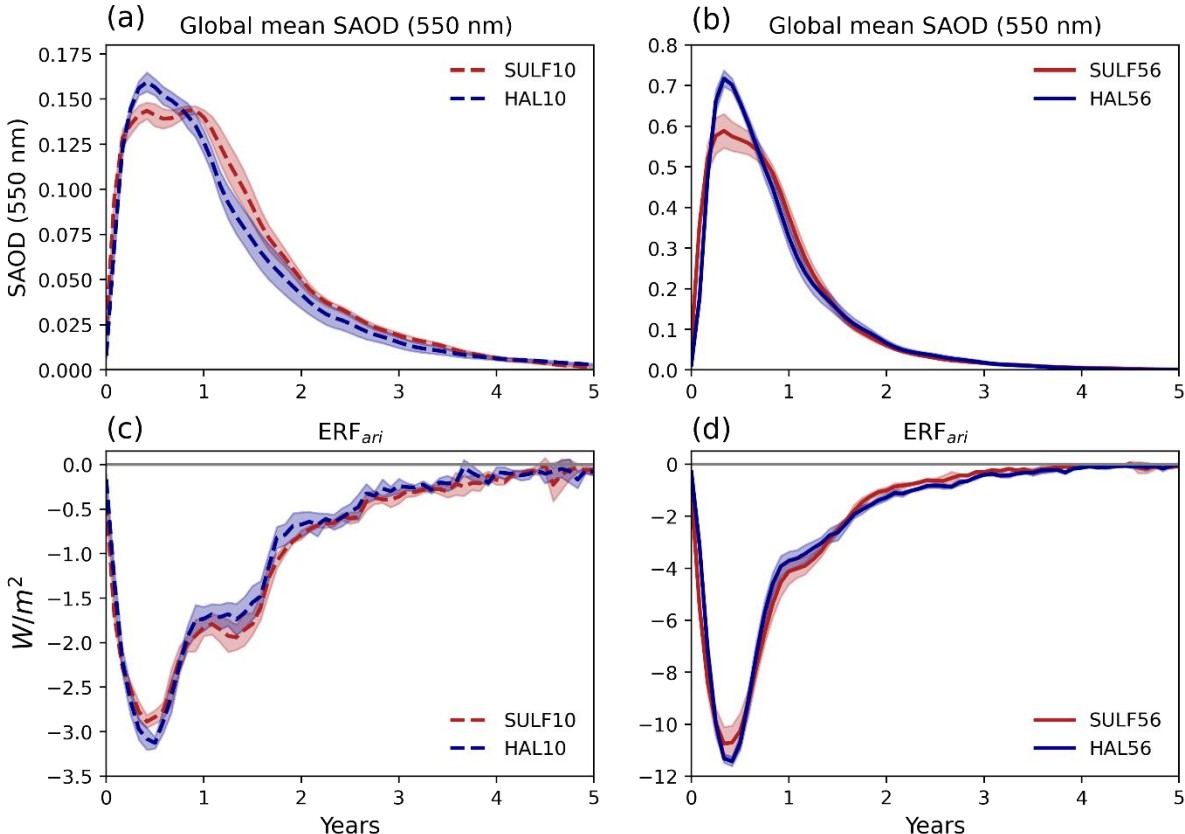

**Figure 4** Global-mean evolution of the stratospheric aerosol optical depth anomaly at 550nm, **(a)** SULF10 and HAL10, **(b)** SULF56 and HAL56. Global-mean evolution of top of atmosphere global-mean ERF$_{ari}$, **(c)** SULF10 and HAL10, **(d)** SULF56 and HAL56. Shading represents the ensemble range.

**3.2 Composition Changes and Resulting ERF$_{clear,clean}$**

Co-emission of volcanic sulfur and halogens causes significant perturbations to the chemistry of the stratosphere beyond the depletion of OH in HAL10 and HAL56 mentioned in section 3.1. Stratospheric methane, stratospheric water vapour (SVW) and, in particular, stratospheric ozone are all impacted.

In sulfur-only simulations, we simulate a modest reduction in global-mean ozone column, -9 DU (-3.9%) in SULF10 and -15 DU (-6.6%) in SULF56 (Figure 5a,c). This ozone depletion is catalysed by halogen radicals activated from background halogens on the surface of volcanic aerosol. We also simulate a redistribution of tropical ozone, with decreases of <0.5 and <2 ppmv between 23 and 28 km and a symmetrical increase in zonal-mean tropical ozone above in SULF10 and SULF56 respectively (Figure 6a,c). This tropical ozone dipole pattern is mostly attributed to volcanic heating. Volcanic heating by the aerosol increases the vertical ascent, and brings ozone up from below enhancing the local mixing ratio. In simulations with co-emitted halogens we simulate more dramatic ozone depletions; HAL10 resulted in a peak global-mean ozone reduction of 65 DU (-22%) 1-2 years after the eruption followed by a gradual recovery over the next 3-4 years (Figure 5d). HAL56 resulted in a peak global-mean ozone reduction of 175 DU (-57%) 1-2 years after the eruption followed by a gradual recovery the

remainder of the 10-year simulation, with an average reduction of 82 DU (-27%) over the 10-year simulation
(Figure 5b).

Volcanic halogen-catalysed ozone depletion is simulated across all latitudes, but the largest magnitude changes in HAL10 (-40%) and HAL56 (-80%) were found within the aerosol cloud and the polar regions, where the co-emitted halogens are activated on aerosol surfaces and PSCs respectively (Figure 5). Ozone depletion
predominantly occurs in the tropics between 25 and 30 km in the first post-eruption year, with depletion maxima of -3.5 ppmv and -6 ppmv in HAL10 and HAL56 respectively (Figure 6). By year three, the ozone depletion shows a similar bimodal altitude distribution in the stratosphere similar to that found in Brenna et al. (2020), with depletion maxima both in the lower (20 km) and upper (40km) stratosphere. As the volcanic $SO_2$ and halogens were introduced into the stratosphere just south of the equator, they are predominantly dispersed into the southern
hemisphere (Figure S6), leading to larger ozone depletions compared with the northern hemisphere. In both HAL10 and HAL56 tropical ozone was found to recover first with significant depletions recurring during the winter in the polar regions for the remainder of the simulation.

The simulated changes in stratospheric heating following sulfur-only and co-emission eruption scenarios affect
the dynamical response of the upper atmosphere, for example, the strength of the Arctic and Antarctic polar vortices (see Figure S7) (Robock, 2000; Toohey et al., 2014). In SULF10 and SULF56, the positive stratospheric temperature anomalies in the tropics lead to an increased meridional temperature gradient. As a result, we simulate a strengthening of the polar vortex (defined as the mean zonal wind speed at the vortex edge, between 55° - 65° latitude and 1 to 30 hPa) in both the Arctic and Antarctic in the first post-eruption winter. In contrast, the negative
stratospheric temperature anomalies in HAL10 and HAL56, lead to a decreased meridional temperature gradient and a weakening of the polar vortices. In HAL10 we simulate significant weakening of the polar vortex in the first two post-eruption winters in the Arctic, and the first and third post-eruption winter at the Antarctic. In HAL56, we simulate significant weakening of the polar vortex for 3-4 years at both poles. Polar vortex strength is an important driver of ozone depletion, with stronger polar vortexes leading to enhanced ozone depletion (Solomon,
1999; Zuev and Savelieva, 2019). Lawrence et al. (2020) linked an unusually strong Arctic polar vortex with the record-breaking ozone loss observed in the 2019/2020 Arctic winter. As such, the strengthening of the polar vortices simulated in sulfur-only simulations may intensify ozone depletion in the first post-eruption winters in both the Arctic and Antarctic. Furthermore, the weakening of the polar vortices simulated in co-emission scenarios may dampen the ozone response in both the Arctic and Antarctic. In addition, the simulated changes in polar
vortex strength may have important consequences for the North Atlantic Oscillation and Southern Annular Mode (Driscoll et al., 2012; Kwon et al., 2020).

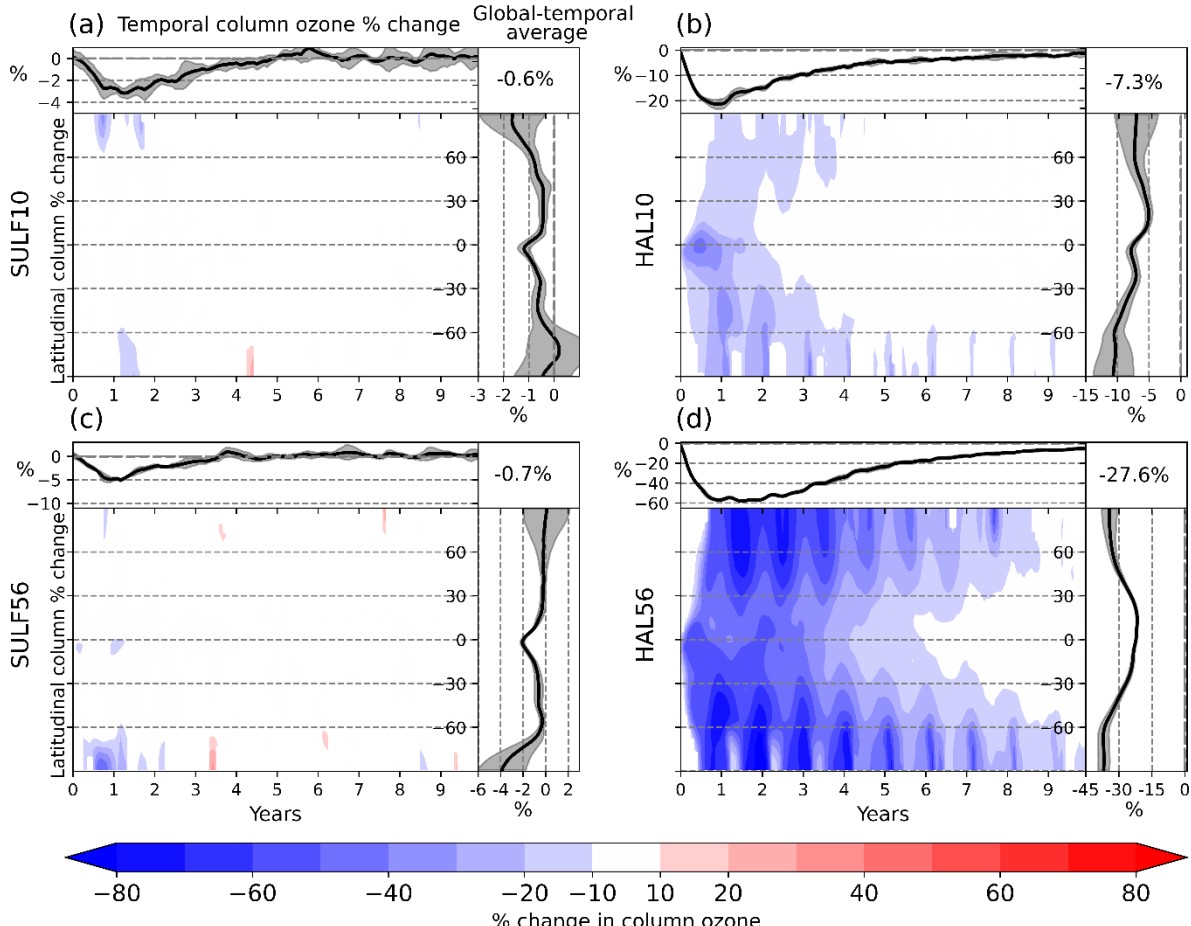

**Figure 5** Ozone percentage difference in **(a)** SULF10, **(b)** HAL10, **(c)** SULF56, **(d)** HAL56. Global-mean total
column ozone anomalies are traced as a function of time at the top of each panel. Temporal-mean ozone anomalies
are traced on the right, note different scales. Global-temporal mean anomalies are enumerated in the top right.
Red colours indicate column ozone enhancement, and blue colours indicate column ozone depletion. Grey shaded
areas represent the ensemble range.

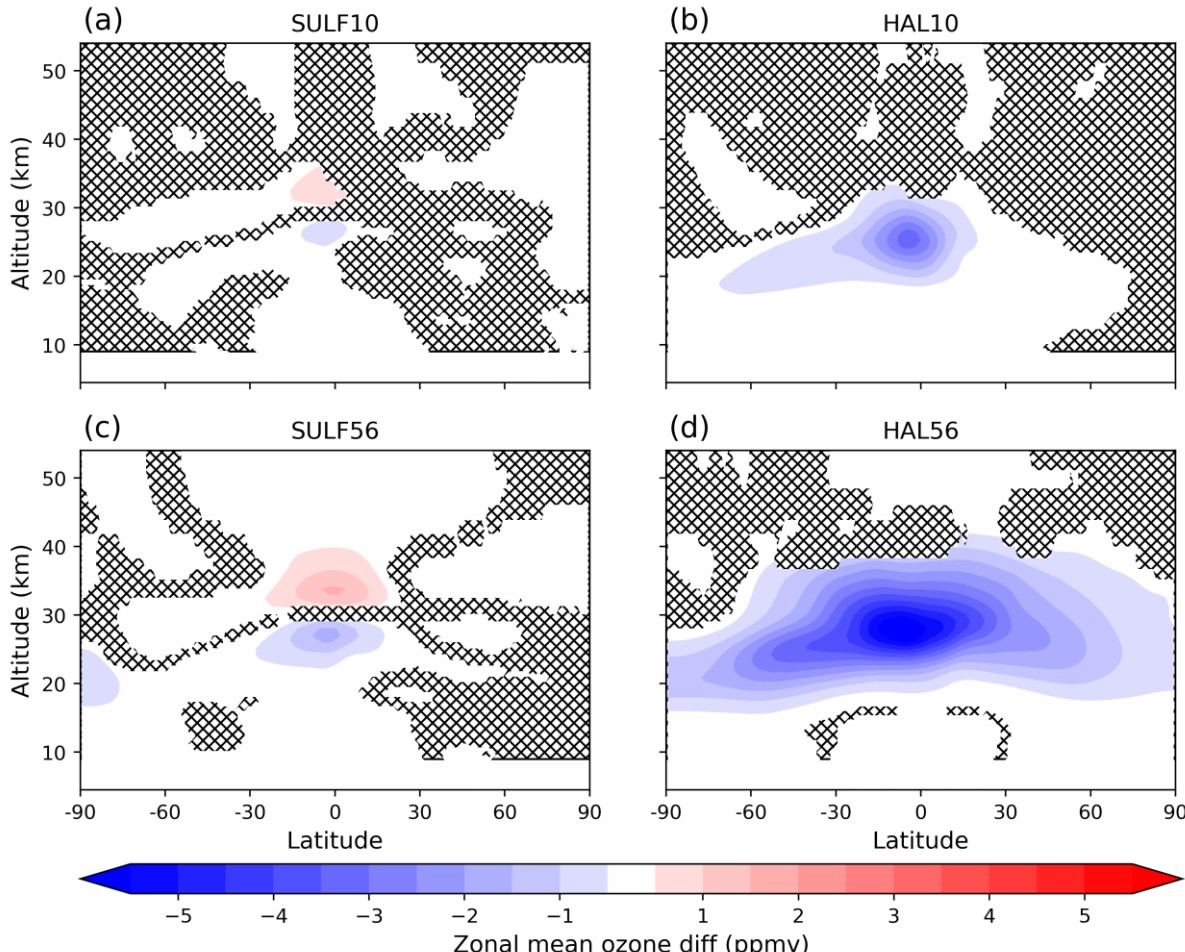

**Figure 6** Zonal-mean ozone anomaly (ppmv) averaged over the first post-eruption year relative to the control climatology, **(a)** SULF10, **(b)** HAL10, **(c)** SULF56, **(d)** HAL56. Red colours indicate ozone enhancement, and blue colours indicate ozone depletion. Anomalies that are not significant at the 95% confidence interval according to a Mann–Whitney U test are indicated with stipples.

Stratospheric water vapour (SWV) and stratospheric methane are linked. SWV has two main sources: transport from the troposphere and chemical production from methane (Löffler et al., 2016). By contrast, stratospheric methane's only source is transport from the tropics and it is destroyed by OH (forming SWV) and reaction with halogens via equation 5.

$$Cl + CH_4 \rightarrow HCl + CH_3 \qquad Eq. 5$$

Following sulfur-only eruptions we simulate small enhancements in SWV and stratospheric methane (Figure 8). SULF10 and SULF56 result in a peak global stratospheric mean increase in SWV of 0.4 ppmv (+7%) and 1.1 ppmv (+17%) and a 10 ppbv (0.6%) and 30 ppbv (1.8%) increase in stratospheric methane respectively. Perturbations to SWV and stratospheric methane peak 2-3 years after the eruption and recover within 7 years. The increase in stratospheric methane following sulfur-only eruptions is in broad agreement with both Loffler (2015), who showed stratospheric methane mixing ratios increased by ~5% following simulations of El Chichón and 15-

20% following the larger Mt Pinatubo, and Kilian et al. (2020) who reported a 10% increase in $CH_4$ between 40 and 10 hPa, also following simulations of Pinatubo. Kilian et al. (2020) suggested that this was due to enhanced vertical ascent as a result of aerosol heating, lifting relatively methane-rich air from the lower stratosphere into the upper levels. As Kilian et al. (2020) simulated an increase in stratospheric $CH_4$ burden, they suggested that the lofting of methane must also coincide with an increase in the stratospheric methane lifetime but did not calculate

this. In SULF10 and SULF56 of this work, we simulate an increase in tropical vertical ascent (shown at 50 hPa in Figure S2), however, we simulate a coinciding reduction in the stratospheric methane lifetime, driven by an increase in methane oxidation by OH and Cl. (Figure S8). This suggests that the increased stratospheric methane burden following sulfur-only eruptions SULF10 and SULF56 is not due to a lengthening of the stratospheric methane lifetime and, instead, is likely due to increased transport of methane across the tropopause from the

methane rich troposphere as a result of increased vertical ascent in the stratosphere (Figure S2). Due to the model set up employed in this study we were unable to diagnose this any further.

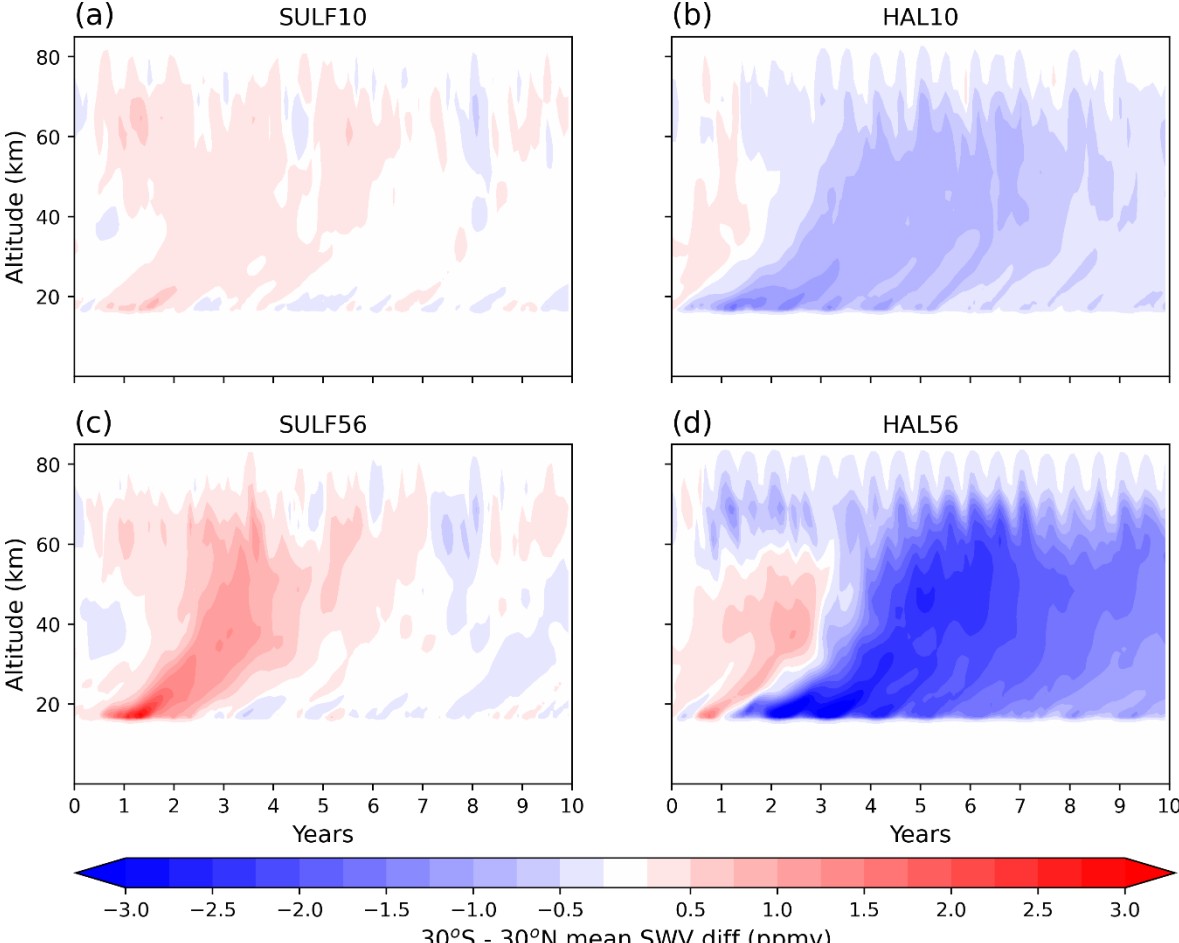

**Figure 7** 30ºS - 30ºN mean stratospheric water vapour anomalies (ppmv) relative to the control climatology as a
function of altitude and time, **(a)** SULF10, **(b)** HAL10, **(c)** SULF56, **(d)** HAL56. Red colours indicate SWV enhancement, and blue colours indicate column SWV depletion.

The simulated changes in methane are small in comparison to the SWV changes across all simulations and can only account for a fraction of the SWV change. The dominant driver of SWV change is the amount of water

vapour entering the stratosphere through the tropical tropopause cold point (Löffler et al., 2016). Following both SULF10 and SULF56, volcanic aerosol results in warming of the tropical tropopause cold point leading to an increase in vertical ascent (Figure S2) and a weakening of the tropical tropopause cold trap dehydration effect, increasing the transport of water vapour into the stratosphere (Figure 8) (Loffler et al., 2016). Elevated SWV is seen to initiate at the tropical troposphere before propagating higher into the stratosphere (Figure 7a,c).


Unlike in sulfur-only eruptions, following eruptions with co-emitted halogens we simulate a reduction in SWV and stratospheric methane (Figure 8). HAL10 and HAL56 result in peak global stratospheric mean stratospheric methane reductions of 37 ppbv (-3%) and 214 ppbv (-18%) respectively 2 years after the eruption. In HAL10 the stratospheric methane perturbation returns to the background levels by the fourth year whereas in HAL56 the

perturbation remains below zero for between 7 and 8 years. Co-emission of halogens results in enhanced destruction of methane by chlorine via Eq. 5 resulting in the significant decrease in the HAL10 and HAL56 stratospheric methane levels.

HAL10 and HAL56 result in peak SWV reductions of 1.0 ppmv (-16%) and 2.3 ppmv (-36%), respectively, 3-4

years after the eruption followed by a gradual recovery. In HAL10 SWV perturbation levels return to the background levels within 7 years whereas in HAL56 the perturbation does not fully recover within the 10-year duration of the simulation. Just as was the case following sulfur-only eruptions, the dominant driver of SWV changes is the amount of water vapour entering the stratosphere via the tropical tropopause cold point. In HAL10 and HAL56, the process is the same but in the opposite sense. Cooling in the tropical tropopause vicinity increases

the efficiency of the tropical cold trap dehydration effect and reduces the amount of water vapour being brought up from the troposphere (Figure 8) (Löffler et al., 2016). The negative SWV anomalies can be seen to initiate at the troposphere before propagating higher into the stratosphere (Figure 7b,d)

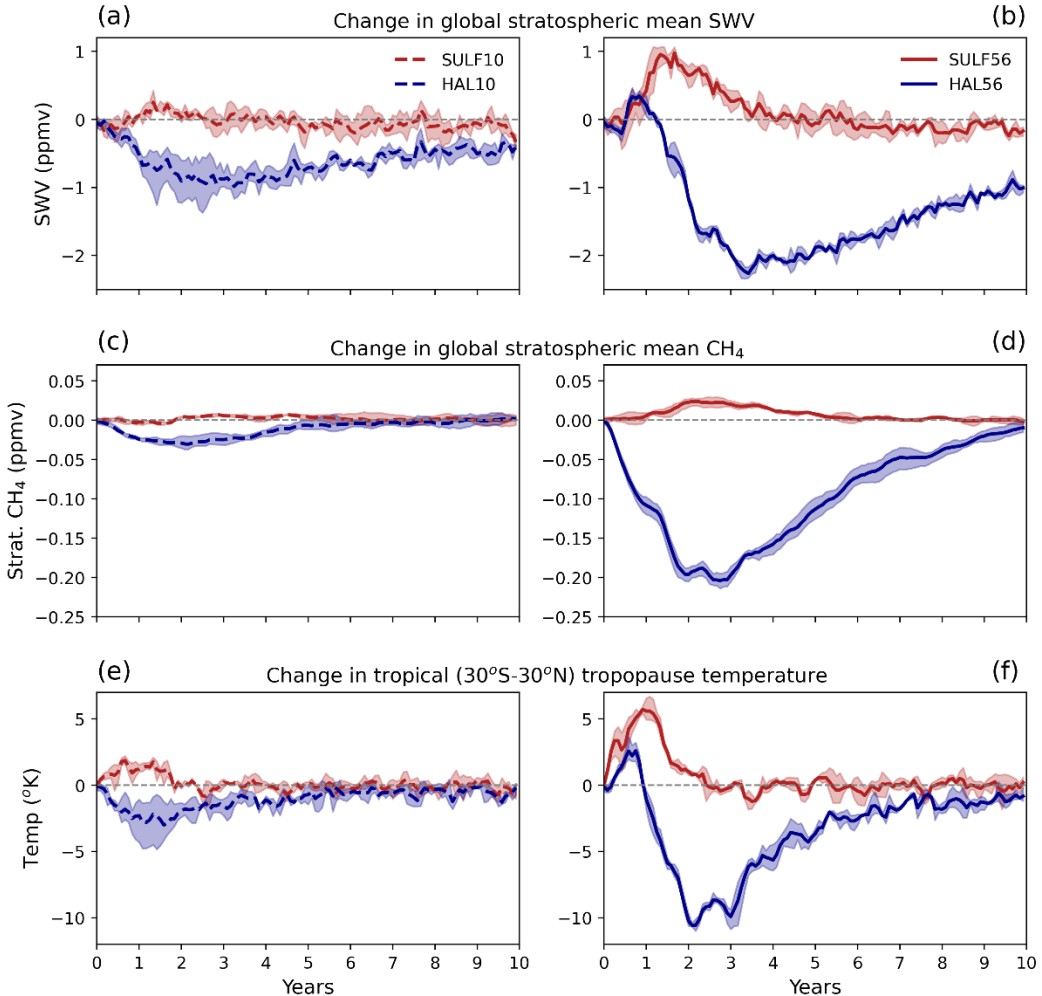

**Figure 8** Evolution of global stratospheric mean water vapour anomalies (ppmv) in SULF10 and HAL10 **(a)**, and SULF56 and HAL56 **(b)**. Evolution of global stratospheric methane anomalies (ppmv) in SULF10 and HAL10 **(c)**, and SULF56 and HAL56 **(d)**. Evolution of tropical tropopause cold trap temperature anomaly averaged over 30ºS–30ºN and 15-20 km in SULF10 and HAL10 **(e)**, and SULF56 and HAL56 **(f)**. Shading represents the ensemble range.


Using the forcing diagnosis outlined in Schmidt et al. (2018) and Ghan (2013), we can isolate the radiative forcing due to atmospheric composition and surface albedo changes, *ERF$_{clear,clean}$*. As surface temperature and sea ice were prescribed, surface albedo changes were small, meaning that ERF$_{clear,clean}$ predominantly represents the forcing from atmospheric composition changes (Figure 9 c, d). HAL10 results in a peak global-mean *ERF$_{clear,clean}$* of -1.3 Wm$^{-2}$ one year after the eruption, more than double the *ERF$_{clear,clean}$* of SULF10. The forcing recovers gradually over the next 6-7 years and results in a cumulative *ERF$_{clear,clean}$* that is 5 times greater than SULF10 (Figure S2d). Similarly, HAL56 results in a peak global-mean ERF$_{clear,clean}$ of -2.1 Wm$^{-2}$ 1-2 years after the eruption, double the peak global-mean forcing of SULF56. The *ERF$_{clear,clean}$* anomaly in HAL56 is more persistent and remains -0.5 Wm$^{-2}$ below zero at the end of the simulation, resulting in a cumulative *ERF$_{clear,clean}$* that is 10 times greater than SULF56 (Figure S2c).



To calculate the resulting radiative forcing from the ozone changes simulated in this work, we use the ozone radiative kernel ($O_3$ RK) technique based on Rap et al. (2015) and updated for the whole atmosphere as outlined in Iglesias-Suarez et al. (2018) (Figure S9). The $O_3$ RK is constructed by calculating the change in LW and SW flux caused by a 1 ppb perturbation in ozone added to each atmospheric layer in turn. The change in SW and LW flux is diagnosed using the offline version of the Suite Of Community RAdiative Transfer (SOCRATES) model, based on Edwards and Slingo (1996). The LW component of the $O_3$ RK (Fig. S9c) is positive throughout the atmosphere, with a maximum in the tropical upper troposphere lower stratosphere. The SW component (Fig. S9c) is negative above ~12 km altitude and positive below ~12 km altitude. This results in a net $O_3$ RK (Fig. S9a) which is positive everywhere except above ~25 km between 60°S and 60°N. Using the $O_3$ RK, we are able to show that the stratospheric ozone change is the dominant driver of the $ERF_{clear,clean}$ accounting for ~75% of the $ERF_{clear,clean}$ (Figure 9a,b). The remainder is likely predominantly due to SWV changes with a small contribution from stratospheric methane changes. The latitudinal pattern of ozone radiative forcing reflects the locations of the ozone change, with largest forcing at the poles, as shown in Figure S10 and S11.

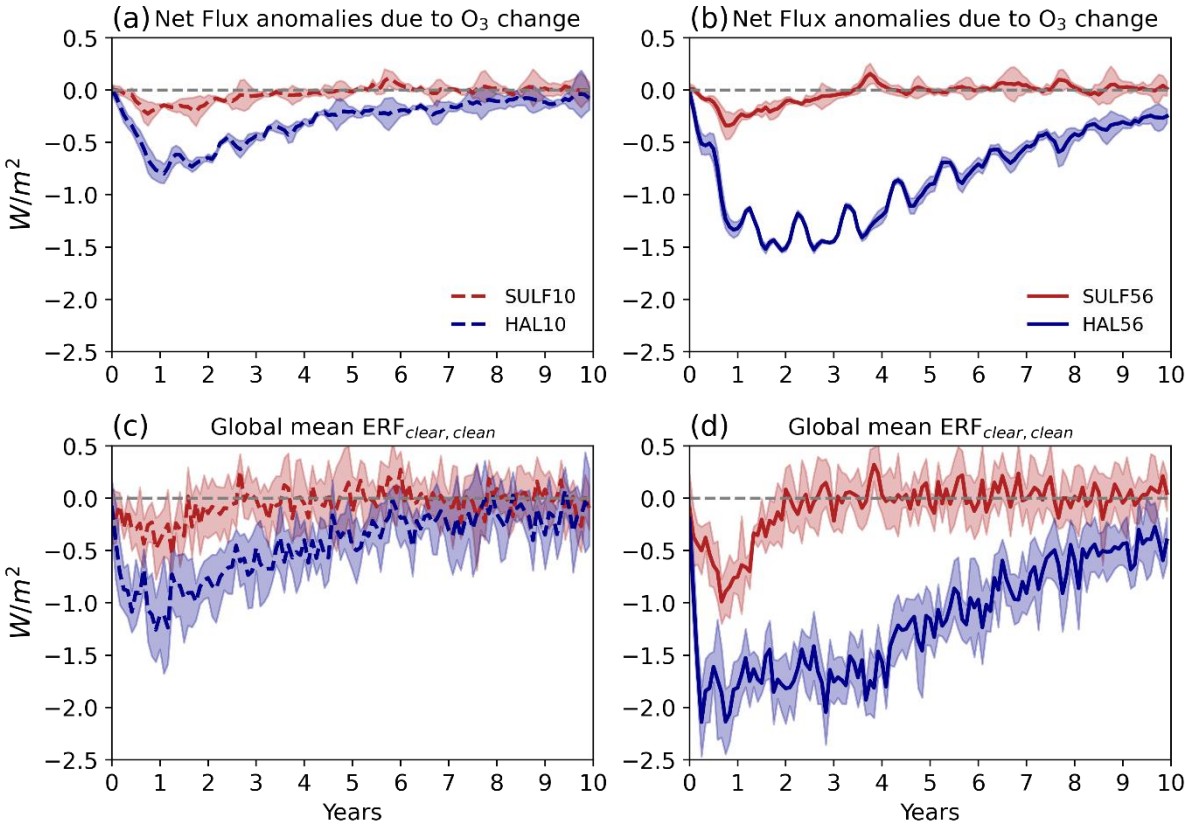

**Figure 9** Evolution of global-mean top of atmosphere net flux anomalies due to stratospheric $O_3$ change estimated from the ozone radiative kernel from Rap et al. (2015) in SULF10 and HAL10 **(a)**, SULF56 and HAL56 **(b)**. Evolution of the global-mean top of atmosphere compositional forcing ($ERF_{clear,clean}$) in SULF10 and HAL10 **(c)**, SULF56 and HAL56 **(d).** Ozone changes make up ~75% of the $ERF_{clear,clean}$. Shading represents the ensemble range.

## 4 Discussion

Using the Ghan (2013) method for diagnosing forcing, we have shown that the co-emission of volcanic halogens

results in larger peak global-mean $ERF_{ari}$ and $ERF_{clear,clean}$. Taking these in combination, the co-emission of halogens results in substantial increases in the peak global-mean volcanic $ERF$ to -4.1 Wm$^{-2}$ (+30%) in HAL10, and -14.1 Wm$^{-2}$ (+24%) in HAL56 (Figure 10a,b), as well as increases in the total cumulative forcing to -1.37 x10$^{23}$ J (+60%) in HAL10 and -3.86x10$^{23}$ J (+100%) in HAL56 compared to SULF10 and SULF56 (Figure S5e,f). In both HAL10 and HAL56, ~25% of the additional peak global-mean volcanic $ERF$ simulated compared to

SULF10 and SULF56 respectively comes from the changes to the $ERF_{ari}$, with the remainder coming from changes to $ERF_{clear,clean}$.

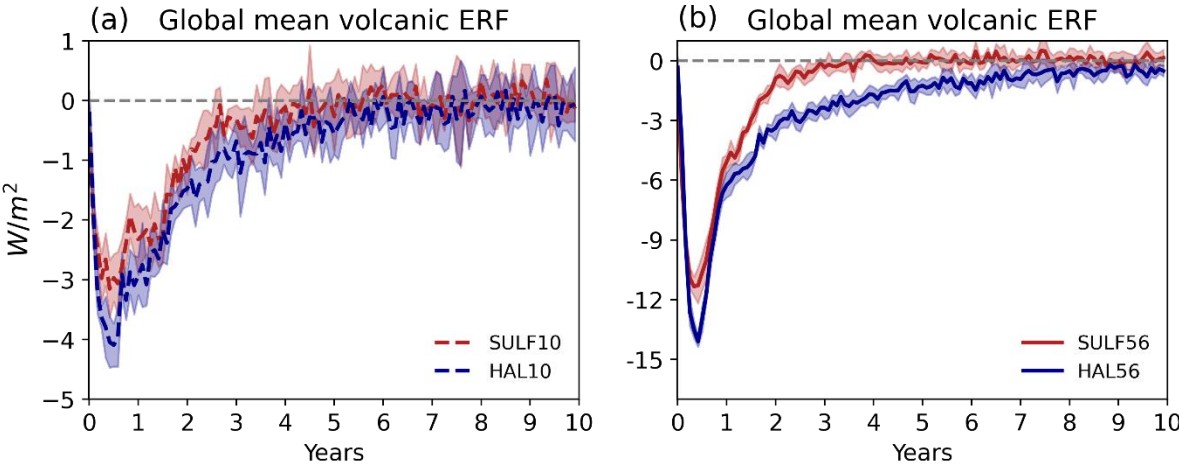

**Figure 10.** Evolution of the global-mean top of atmosphere total volcanic forcing (*ERF*) relative to the control climatology in SULF10 and HAL10 **(a)**, SULF56 and HAL56 **(b).** Volcanic *ERF* is the sum of $ERF_{ari}$, $ERF_{acii}$

and $ERF_{clear,clean}$. Shading represents the ensemble range.

Comparing the perturbations in HAL56 to HAL10, we find that increasing the volcanic halogen flux by 10 times only results in a ~2.5 times larger global ozone response and, as $ERF_{clear,clean}$ is dominated by changes in stratospheric ozone, only a ~2 times larger $ERF_{clear,clean}$. This suggests that there is a saturation in the ozone

depleting potential of co-emitted volcanic halogens. Plotting the column ozone percentage change against the magnitude of injected halogens expressed as Equivalent Effective Stratospheric Chlorine (EESC is a measure of the ozone destruction potential; EESC=[Cl]$_{added\ to\ stratosphere}$ + 60 x [Br]$_{added\ to\ stratosphere}$; Cadoux et al., 2015) from this study and a number of previous studies, we find an exponential decay curve describes this relationship: as the EESC increases the efficiency of volcanic halogen ozone depletion decreases (Figure 11). This relationship

suggests that column ozone is most sensitive to volcanic halogens when the additional EESC is < 20 Tg, and that increasing the volcanic EESC flux beyond 60 Tg has little impact on column ozone change. This analysis spans simulations with very different background EESC and column ozone values. Wade et al. (2020), Brenna et al. (2019), and Brenna et al. (2020) simulations are all in a pre-industrial atmosphere background states with low background chlorine levels, whereas, the background chlorine levels in HAL10 and HAL56 are significantly

higher and with lower initial ozone columns. This relationship suggests that the peak global-mean ozone loss (%) is dependent more on the volcanically injected EESC than the background chlorine and initial ozone columns. In other words, this relationship is time-independent and this exponential decay curve can be used to estimate the

peak global-mean ozone loss for an eruption in any climate state, including future eruptions where the background EESC will have decayed back to pre-1980s levels. This will be especially useful for rapid estimates of ozone 555 change as new or better constrained volcanic halogen data becomes available.

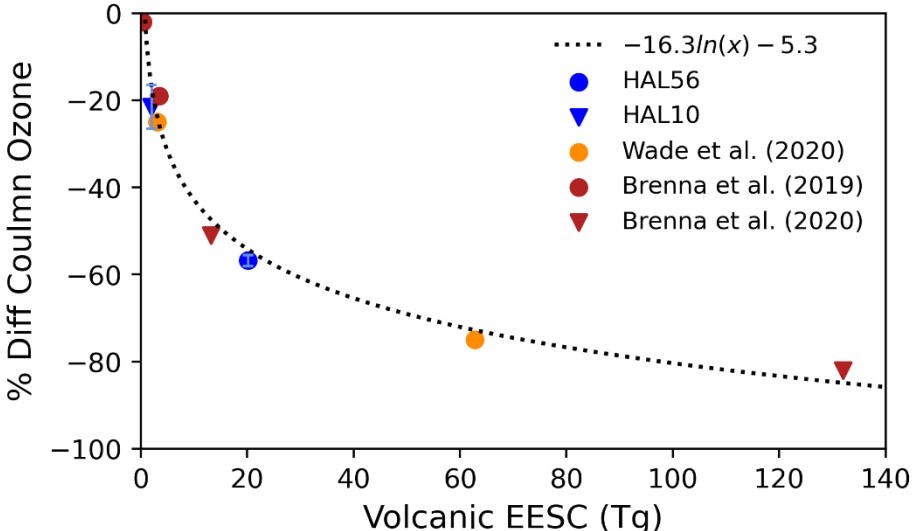

**Figure 11** Relationship between volcanically emitted Equivalent Effective Stratospheric Chlorine (Tg) and peak 560 global-mean % difference in column ozone. Blue: HAL10 and HAL56 ensemble mean and range. Orange: Wade et al. (2020) ensemble mean. Red: Brenna et al. (2019) and Brenna et al. (2020) ensemble mean.

The implications of ozone depletion in HAL10 and HAL56 go further than enhancing the $ERF_{clear,clean}$. High anthropogenic fluxes of halocarbons into the atmosphere during the 1980s caused background chlorine levels to 565 be elevated during the 1990s and an ozone hole is simulated to develop in the control simulation over the southern hemisphere polar regions (Figure S12). Using the definition for ozone hole conditions as <220 DU, we simulate enhanced ozone hole conditions following both HAL10 and HAL56 eruptions (Figure 12). In HAL10, ozone hole conditions are simulated in the tropics for one year after the eruption, and a deepening of ozone hole conditions is seen in northern hemisphere polar regions for two winters and in the southern hemisphere polar regions for four 570 winters. In HAL56, we simulate ozone hole conditions globally for 5 years, continuing for a further three winters in the northern hemisphere polar regions and six winters in the southern hemisphere polar regions

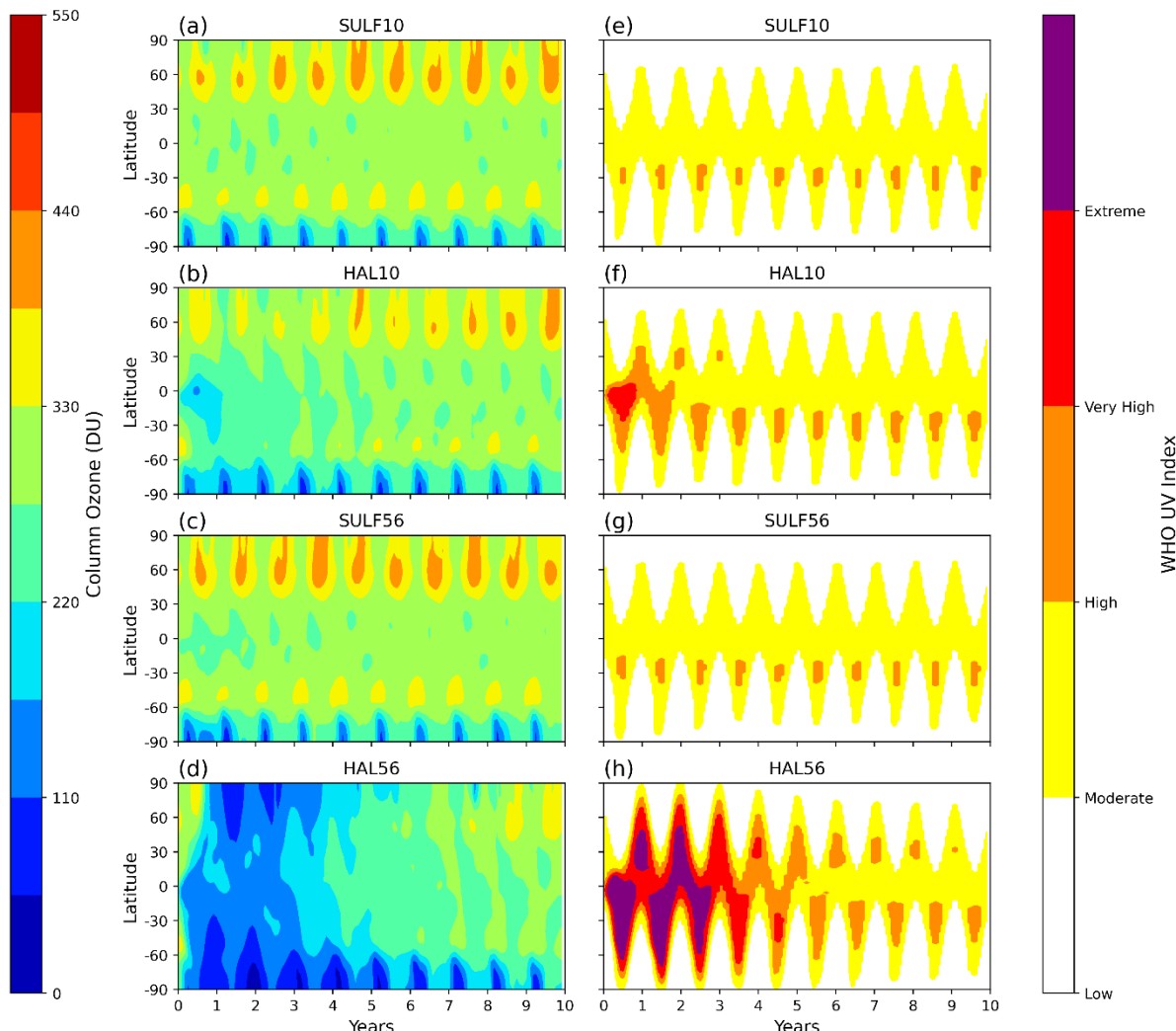

**Figure 12** Zonal-mean column ozone **(a)** SULF10, **(b)** HAL10, **(c)** SULF56, **(d)** HAL56. Ozone hole conditions are simulated when the column ozone <220 DU. Zonal-mean surface UV exposure due to column ozone changes (WHO UV Index) **(e)** SULF10, **(f)** HAL10, **(g)** SULF56, **(h)** HAL56.

Column ozone depletion on this scale would dramatically increase the flux of harmful UV to the surface, which could cause DNA damage to animals and plants, and increase the occurrences of skin cancers, eye damage and immune system deficiencies among the population (WHO, 1994). Climate modelling and environmental proxies showed that ozone depletion as a result of halogen degassing during the emplacement of Siberian Traps flood basalts led to ozone depletion that stressed ecosystems and caused DNA mutations which may have contributed to the end-Permian mass extinction (Black et al., 2014). A simple heuristic relating column ozone to clear-sky surface UV index is given by:

$$UV\ Index\ =\ 12.5\mu_o^{2.42}(\Omega/300)^{-1.23}$$

as defined in (Madronich, 2007), where $\mu_o$ is the cosine of the solar zenith angle and $\Omega$ is the total vertical ozone column in Dobson units. The monthly mean average UV index coloured by World Health Organization categories (Low [0 to 2], Medium [3 to 5], High [6 to 7], Very High [8 to 10], and Extreme [11+]) is shown in Figure 12. This shows that in the HAL56 scenario, on average 'Very High' or 'Extreme' UV levels would be expected all day for much of the globe in the three to four summers after the eruption, with noon values being even higher. The change in surface UV levels are shown in Figure S13. Living under such a high UV exposure would cause immediate immunosuppression, epidemic outbreaks, increases in the occurrences of eye damage and, in the longer term, skin cancers among the population living between the equator and the mid-latitudes, which equates to >95% of the global population. The assessment of surface UV changes is made more challenging by the presence of volcanic aerosols, which also scatter UV radiation. However, damaging UVB and UVC radiation will not be scattered effectively by larger aerosol size distributions and volcanic aerosol levels reduce rapidly after peaking in the first post-eruption year.

Whilst we have been able to calculate the composition and climate impacts of the co-emission of halogens and $SO_2$ from volcanic eruptions, these calculations are not without some uncertainty. Recent studies carried out as part of the Volcanic Forcings Model Intercomparison Project (VolMIP) showed large model response disparities in simulations of $SO_2$-only volcanic eruptions (Clyne et al., 2021), but models have been shown to capture the effects of ozone depleting substances on stratospheric ozone well (World Meteorological Organization (WMO), 2014). As outlined in the introduction, the major uncertainty in this work is the stratospheric injection of HCl and HBr from explosive volcanic eruptions, which is highly variable and depends on both the geochemistry of the volcano and the degree of scavenging determined by the prevailing atmospheric conditions during the eruption. It is clear, however, that significant stratospheric halogen fluxes occur after some explosive volcanic eruptions.

Although this work has focused on simulations of explosive volcanic eruptions in a background climate representative of the 1990s, Figure 11 demonstrates the simulated ozone depletion predominantly depends on the volcanic halogen injection size and not the background atmospheric state. Using the relationship outlined in Figure 11, we can estimate the peak global-mean ozone percentage loss for any size of volcanic halogen injection, past or present. We are currently investigating the impacts that plausible future background atmospheric states (such as different greenhouse gas concentrations, background halogen levels and stratospheric temperatures) may have on the simulated ozone response and volcanic *ERF* due to co-emitted sulfur and halogen volcanic emissions.

In addition to the co-emission of volcanic halogens, there is also scope to model the co-emission of volcanic water vapour and ash directly into the stratosphere. Legrande et al. (2016) provide a mechanism explaining how SWV originating from volcanic eruptions may alter the chemistry of the stratosphere and the nucleation rate of sulfate aerosol and suggest that this may severely alter the climate impacts. In addition, SVW proved to be an amplifying feedback in simulations in this work and it would be interesting to see how co-emission of water vapour, halogens and sulfur would further alter the volcanic forcing in simulations of explosive volcanic eruptions. Zhu et al. (2020) showed the importance of including volcanic ash injections in climate simulations. When heterogeneous chemistry on ash particles was included they found that 43% more volcanic sulfur was removed from the stratosphere in the

first 2 months. Volcanic ash is also likely to alter the lifetime, activation and impact of co-emitted volcanic halogens in climate simulations.

**5 Conclusions**

In this study we utilised UKESM-AMIP simulations of volcanic eruptions to investigate how the co-emission of volcanic halogens and sulfur alters the effective radiative forcing (ERF) of explosive volcanic eruptions under atmospheric conditions representative of the mid-1990s. As the volcanic flux of HCl and HBr into the stratosphere remains uncertain, a range of plausible explosive volcanic emissions scenarios based on petrological degassing

estimates, satellite observations and volcanic plume modelling were simulated. The four sets of experiments included one large $SO_2$ (10 Tg), and one very-large $SO_2$ (56Tg) emission scenario, both with (HAL10 and HAL56) and without halogens (SULF10 and SULF56), each with an ensemble size of 6 sampling different QBO states. These eruption sizes (10 and 56 Tg $SO_2$) are hypothetical, but they are comparable to a VEI 6 (e.g. 1991 Mt. Pinatubo) and a VEI 7 (e.g. 1257 Mt. Samalas) eruption, representing 1 in 50-100 year and 1 in 500-1000 year

events respectively. HAL56 utilises the 1257 Mt. Samalas HCl and HBr emission estimates from Vidal et al. (2016) and assumes a conservative ~5% stratospheric halogen injection efficiency. HAL10 has a SO2 injection similar to that found to reproduce the spatial and temporal evolution of SAOD following 1991 Pinatubo (Mills et al., 2016) and a 10 times smaller HCl and HBr flux than HAL56.

We have shown that the co-emission of halogens and sulfur in simulations of explosive volcanic eruptions significantly increases the peak and cumulative volcanic *ERF*. This is due to a combination of increased forcing from i) volcanic aerosol-radiation interactions (*ERF$_{ari}$*) and ii) composition of the stratosphere (*ERF$_{clear,clean}$*).

Co-emitting halogens results in a larger global-mean *ERF$_{ari}$* in both HAL10 (+8%) and HAL56 (+6%). Ozone

depletion catalysed by volcanic halogens leads to stratospheric cooling which offsets the volcanic aerosol heating (SULF10 $\simeq$ 1.5 K, SULF56 $\simeq$ 3.5 K) and results in a net stratospheric cooling (HAL10 $\simeq$ -2 K, HAL56 $\simeq$ -3.5 K). The ozone-induced stratospheric cooling prevents aerosol self-lofting and keeps the volcanic aerosol lower in the stratosphere with a shorter lifetime, resulting in reduced growth via condensation and coagulation and smaller peak global-mean effective radius compared to sulfur-only simulations. The peak global-mean effective radii of

the HAL10 and HAL56 sulfate aerosols are found to be 15% and 10% smaller than SULF10 and SULF56 sulfate aerosols, closer to the most efficient radii for scattering short wave radiation per unit mass, ~0.20 μm. Subsequently, we find HAL10 and HAL56 have higher peak global-mean SAOD anomalies (+11%, +22%) and *ERF$_{ari}$* (+8% + 6%).

Co-emission of halogens also results in significant perturbations to the stratospheric chemistry and compositional-driven forcing. Stratospheric methane was found to decrease by 3% and 18% and stratospheric water vapour (SWV) was found to reduce by 16% and 36% in HAL10 and HAL56 respectively. The methane reductions were driven by the enhanced destruction flux by volcanic Cl radicals and the SWV changes were attributed to the same stratospheric temperature reductions mentioned previously. Cooling in the tropical tropopause vicinity increased

the efficiency of the tropical cold trap dehydration effect, reducing the flux of water vapour from the troposphere to the stratosphere. The most dramatic change in chemistry was found to be in stratospheric ozone. Significant

ozone depletions were simulated globally in both HAL10 (22%) and HAL56 (57%) with prolonged depletion in both NH and SH winter polar regions. In HAL10, ozone hole conditions (<220 DU) were simulated globally for the first post-eruption year and then for 3-5 years at the poles during the winter. In HAL56, we simulate an ozone

hole globally for 5 years followed by a gradual recovery over the following 5 years until only the polar winters exhibit ozone hole conditions. Stratospheric chemistry changes resulting from the co-emission of halogens increased the peak global-mean $ERF_{clear,clean}$ by ~100% to -2.1 Wm$^{-2}$ in HAL56 and -1.3 Wm$^{-2}$ in HAL10. Stratospheric ozone depletion is the dominant driver of $ERF_{clear,clean}$ accounting for ~75% of the total $ERF_{clear,clean}$.

The combined effect of increased $ERF_{ari}$ and $ERF_{clear,clean}$ is that co-emitting halogens increases the peak global-mean volcanic $ERF$ by 30% and 24% and cumulative $ERF$ by 60% and 100% in HAL10 and HAL56 respectively. Ozone hole conditions exhibited by both HAL10 and HAL56 would result in dramatic increases in the surface UV flux with 'Extreme' UV levels being experienced over most of the globe for 4 years following HAL56 eruptions. UV exposure on this scale would lead to devastating negative consequences for society and the

biosphere, including increases in the occurrences of skin cancer, eye damage and immune system deficiencies (WHO, 2002). This work shows for the first time that co-emission of plausible amounts of halogens can amplify the effective radiative forcing in simulations of explosive volcanic eruptions. This highlights the necessity to include volcanic halogens emissions when simulating the climate impacts of past or future eruptions, and the critical need to maintain space-borne observations of stratospheric compounds to better constrain the stratospheric

injection estimates of volcanic eruptions.

**Data Availability**

All data required to reproduce our key results are archived in the Centre for Environmental Data Analysis (CEDA) archive, and can be found here: https://catalogue.ceda.ac.uk/uuid/5f4d2f6daebd4195a0368a79405d3686. Post-processing and visualization of data was performed with Python. The scripts and the post-processed data files are

available on request from the corresponding author.

**Author Contributions**

J.SS designed the study, ran the UKESM1-AMIP experiments, analyzed the results and wrote the manuscript. A.S. and A.A. provided support for designing the study and analyzing the results. T.A., Y.M.S, J.W., L.R.M., N.L.A. provided support for running the experiments, and T.A., Y.M.S, J.W., provided support for the analysis.

All authors contributed to revising the manuscript.

**Competing interests.**

The authors declare that they have no conflict of interests.

**Acknowledgements**

JSS and YMS would like to thank NERC through the University of Cambridge ESS-DTP for funding; JW would
like to thank the Cambridge Commonwealth, European & International Trust for funding through a Vice
Chancellor's Award. T.J.A. acknowledges support from the Royal Society through a Newton International
Fellowship (grant number NIF\R1\180809), from the European Union's Horizon 2020 research and innovation
programme under the Marie Skłodowska-Curie grant agreement No 835939, and from the Sidney Sussex college
through a Junior Research Fellowship. L.R.M. and A.S. are funded by the U.K. Natural Environment Research
Council (NERC) via the "Vol-Clim" grant (NE/S000887/1). In addition, A.S. acknowledges funding via the
NERC V-PLUS project (NE/S00436X/1). We would like to thank NERC, through NCAS, and the Met Office for
the support of the JWCRP UKCA project. NLA and ATA are supported by NERC and NCAS through the ACSIS
project. The team thank NCAS and the Met Office, through the JWCRP, for support of the UKCA model. This
work used Monsoon2, a collaborative High Performance Computing facility funded by the Met Office and the
Natural Environment Research Council. This work used JASMIN, the UK collaborative data analysis facility. We
would like to thank Alex Rap for allowing the use of the ozone radiative kernel and for his help implementing it
in this work. Finally, we thank Daniele Visioni, an anonymous reviewer, and Alan Robock whose useful and
constructive comments helped to improve and clarify this manuscript.

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
