# Peer review of "Co-emission of volcanic sulfur and halogens amplifies volcanic effective radiative forcing"

_Atmospheric Chemistry and Physics, 2020_

## Referee Comment (RC1) · Daniele Visioni (Referee) · 18 Nov 2020

In this study the authors, using a CCM model, try to understand how co-emitted halogens may alter the climate impacts of stratospheric sulfur injection by volcanic eruptions. Their results show that the inclusion of halogens dramatically changes some of the simulated impacts, and that including such emissions is crucial to properly simulate large explosive volcanic eruptions. I found the paper to be excellent: the introduction does a great job at framing the problem and the methods are properly described. The result are also explained clearly, with pretty straightforward figures (which I suggest uploading in a higher quality format, they look a bit blurred when zoomed in). Overall, the manuscript is perfectly suitable to be published in ACP.

[Figure]

I have only a few minor comments:

L 69: highlight with no s L 93: The correct name of the model is CESM1(WACCM) (Community Earth-System Model and then WACCM) L 116: from the way the phrase is written, it looks like the different halogen emissions are applied to the same amount of SO2, but in that case, the ratio would also have to be 100 times. But Ming et al. 2020 compare also a low and high SO2 injection (10 vs. 100 Tg) and in there different HCl concentrations. Just try to clarify this point.

Fig. 1: please specify what is the quiescent period against which the anomalies are calculated in the caption. Fig. 1f: I'm a bit confused as to why in the SULF simulations, there is a small increase in OH that I don't think is properly explained in the text. In the SO2 plume, we expect a large OH depletion. I assume that can be balanced out by the influx of water vapor in the stratosphere from the lower stratospheric heating and produce globally a slight increase. But I'd suggest checking (or consider the tropical changes in stratospheric OH, where I'm sure the change is negative – albeit less than in the HAL experiments).

Line 255: it would be useful to show the changes in w* (maybe in the supplementary next to Fig. S1) to show the difference in the transport induced by the stratospheric heating.

Fig. 3: I'd suggest switching panels a and b, as logically one might expect the lower injection scenario before. Also, I find it interesting that the relationship doesn't hold as well for the lower injection case. I suspect this might be due to different QBO phases that affect the aerosols e-folding time (see Pitari et al., 2016), and that this effect is more evident for lower injection rates while for higher injection rates the increase heating rates modify the QBO too strongly independently on the phase it's in at the moment of injection (see for instance Aquila et al., 2014) resulting in similar lifetimes. The authors could just check if that's the case verifying the QBO phase, or just mention that's a possibility for the lower correlation in panel b (unless they have a better explanation).

Fig. 8: please specify if panels a-d are global changes.

Aquila, V., Garfinkel, C. I., Newman, P. A., Oman, L. D., and Waugh, D. W. (2014), Modifications of the quasi‐biennial oscillation by a geoengineering perturbation of the stratospheric aerosol layer, Geophys. Res. Lett., 41, 1738– 1744, doi:10.1002/2013GL058818.

Pitari, G., Genova, G. D., Mancini, E., Visioni, D., Gandolfi, I., & Cionni, I. (2016). Stratospheric aerosols from major volcanic eruptions: A composition-climate model study of the aerosol cloud dispersal and e-folding time. Atmosphere. https://doi.org/10.3390/atmos7060075

---

## Referee Comment (RC2) · Anonymous Referee #2 · 16 Dec 2020

This study considers hypothetical VEI 6 and VEI 7 sized eruptions and, using a coupled chemistry-aerosol model driven by scenarios including or not including halogen injection, investigate how the co-emission of volcanic sulfur and halogens alters the evolution of the volcanic aerosol plume, stratospheric ozone chemistry, and the resulting radiative forcing and UV flux.

The authors investigate how volcanic halogens may interact with the sulfur aerosol life cycle and interact to modulate volcanic forcing conversely to previously reported work. I found the link between chemical and microphysical processes of particular interest. The authors reveal in their model experiments the primary importance of halogens in major volcanic emissions in the sulfur cycle in the stratosphere, a process already suspected for eruptions of much more minor amplitude. Impacts of halogen emissions

on dynamics of the aerosols and the subsequent effect on aerosol microphysics are also considered here when critically missing in previous reported studies. Effects on some key stratospheric compounds like ozone, water vapour and methane are also analysed. To me, this work points at the critical need to maintain space-borne observations of stratospheric compounds which will be particularly valuable to quantify halogen injected by volcanoes and needed for model initialization. The authors finally mention open questions to be addressed in future studies reflecting the great interest to consider these events and their associated various injections (sulfur, halogens, ash, water vapour) in the future climate.

This study is original and comprehensive judging by the various topics and impacts covered (microphysics, dynamics, chemistry and radiative forcing). I found the manuscript clear, well-written and nicely going straight to the obtained results. I estimate that this work deserves to be published in ACP after the following minor comments have been addressed.

Specific comments:

Introduction: I am not a specialist of petrological processes but could you indicate the degree of uncertainty when petrological budgets are used to derive stratospheric inputs of halogens? What are the assumptions behind the halogen injection efficiency (I would suggest to briefly recall the definition of the stratospheric halogen injection efficiency).

P5 lines 158-168: How the overall chemical species initialized? Is it based on climatological 3D fields or only surface emissions provided by CMIP6? Especially for bromine compounds, how the $Br_y$ budget initialized? Are very-short-lived species accounted for? The resulting inorganic bromine budget in the stratosphere and more generally the inorganic halogen content, computed in chemistry models is of particular importance regarding ozone chemical cycles and would have significant impact in your scenario with no volcanic emissions of halogens. Please provide a bit more information.

P5 line 170 onwards: You do not differentiate between SW and LW radiation conversely to the work of Schmidt et al. (2018). What justifies this choice?

P5 line 188: why variations in the surface albedo were not taken into account? As you state in the manuscript, the model can be forced by surface boundary conditions. Is it for calculation-time issues?

P6 line 202: An injection altitude distributed around 21 km has been chosen. Volcanic impact depends on injection altitude especially because the residence time of aerosols is linked to this parameter. I am aware that strict choices must be done for costly long-term simulations but what justifies this value? Did the authors conduct sensitivity tests on this parameter?

P6 lines 206-219: there is a lot of assumption behind the stratospheric halogen injection efficiency. The values given by Textor et al. (2013) strongly differ from other reported studies. Is this factor highly variable from one eruption to another? Why El Chichon and Mazama eruptions reassures the numbers taken for HAL56? For HAL10 it is not clear to me why the Pinatubo HCL:SO2 molar ration must match the one for Mt Mazama. I guess petrological processes somewhat differ for these events. Please clarify.

P8 figure 1d: what is the reason for the overlapping HAL and SULF S global burden at an early stage of the simulations? This surprises me because this feature is not visible on SO4 which already shows a marked difference over the first months (figure 1c).

P11 lines 268-279: The investigation of Reff is interesting since it provides a (integrated) description of the impact of HAL scenario on aerosol sizes. However, it would have been also valuable to examine more comprehensively the impact on microphysics. Although GLOMAP is a modal microphysical module (as far as I understood) did the authors get information about the effects on size distributions (geometrical standard deviation, total concentration)? For instance, concentrations might be reduced if particle sizes increase but with different ratios. Concentration (well, the whole size distribution) is also important for the SAOD and ERF calculated in the manuscript (figure 4).

P11 lines 268-279: I think a short comparison with maximum Reff values reported for different past eruptions (Pinatubo in particular) would be interesting to include here.

P11 line 291: please add the wavelength here.

P12 figure 4: the color coding (red for SULF and blue for HAL) is the opposite than in previous figures. I think it would be preferable to homogenise this.

P14 figure 6: A latitudinal transport is visible in the lower stratosphere, particularly for HAL10 simulation. What process can be related to the hemispherical difference? Since the plot is integrated over 3-years I am not sure that the dominating phase of the QBO (which has been shown to impact volcanic aerosol transport from the tropics) can be an explanation.

P15 lines 360-366: I guess the methane increase, although limited, for the sulfur-only scenario is chemically due to the reaction of $CH_4 + OH$. This would mean that less OH is present under volcanically-impacted periods. There is a complex interplay between $HO_x$, nitrogen and halogen chemistry that can result in OH reduction (and $CH_4$ increase) unless the dominating process deals with the very high amounts of $SO_2$ that may sequester OH through reaction $SO_2+OH$ (subsequently leading to the formation of sulfuric acid). In my mind, OH was rather increased for summertime midlatitude eruptions (as shown for the 2009 Sarychev eruption) reflecting a possible seasonal effect. Changes in methane amounts are also likely resulting from radiative/dynamical origin with more troposphere-to-stratosphere transport resulting from the aerosol heating in the tropopause region. A significant part of ozone changes following major eruption has been attributed to changes in transport (see e.g. Pitari, G. and Rizi, V.: An estimate of the chemical and radiative perturbation of stratospheric ozone following the eruption of Mt. Pinatubo, J. Atmos. Sci., 50, 3260–3276, 1993). Similar process could apply for methane. Do the authors have an idea about the process behind the $CH_4$

slight increase?

P18 line 404: I found the demonstration about ozone change (although largely trusted) as the dominant driver of ERFclear, clean a bit abrupt. Please provide more details here about the method used rather than only citing the Rap et al. reference.

P20 line 461: specify "for two winters".

P21 figure 12: this figure (as figure S4) is very interesting but I think plotting anomalies (by subtracting each simulation with the control run) would have been more meaningful especially to highlight the effect of SULF and HAL scenarios on the NH high latitudes. Such figure could be added in the supplementary material.

---

## Short Comment (SC1) · 28 Dec 2020

I would recommend to the Editor major revisions. Here are the main issues:

1. I don't understand how you carried out your simulations, and why you did them the way you did. Did you take an 11-year average of SSTs and sea ice, and then prescribe them, including their seasonal cycle, repeating the same average year for each entire simulation? Why 11 years? Do you understand that this removes all interannual variation in SSTs, and removes many surface feedbacks with the climate system? The 11-year period you are choosing, 1990-2000, includes the 1991 Pinatubo eruption. Was it included in the forcing for the coupled GCM you used? How did that affect the climate, and why are you averaging over its impacts. Furthermore there were a

moderate and a huge El Niño in that decade. Did the GCM simulate them? Does your SST pattern have a permanent El Niño? How does that affect the climate response? And why did you use the Vidal et al. emissions for Cl and Br, but not for SO2? They said Samalas emitted 158 Tg, but you several times say 56 Tg is representative of the Samalas eruption.

2. You need, in the introduction, an explanation of the ways volcanic eruptions affect stratospheric ozone, including before there were anthropogenic CFCs there and now. Since you are a chemist, chemical reactions would be useful. And dynamic processes also should be explained.

3. I see no discussion of the impacts of volcanic eruptions on stratospheric dynamics. How do changes in stratospheric circulation affect the ozone distribution and the aerosols? You say something about a lower branch of the BDC, but do not show what you are talking about, and how the circulation changes in response to the volcanic eruptions. How does the polar vortex respond, and how does this affect the size of the Ozone Hole?

4. All the time plots need an x-axis in years and not months, so that the seasonal cycles are easy to discern. Months since July 1 are confusing and obscure what is happening. You can start all your plots on January 1 of the year you injected the gases, so we can see what the variability was before the experiment started, too. The latitude plots need labels of 0°, 30°, 60°, and 90°, not 50 only. Climate scientists are used to looking at the different regions of Earth on the natural coordinates.

5. You use many acronyms without defining them. And you define some acronyms more than once.

6. All variables need to be in italics, and chemical symbols should not be in italics.

7. You use r for both correlation and radius, even in the same figure, which is very confusing. You can only use a symbol for one thing in a paper.

8. You use a lot of global averages without showing the spatial and temporal patterns. A lot of what is happening depends on location and time of year. The specific processes are what is of interest. You can average the final result afterwards, but what is important is why things change and where and what time of year they change. This affects the climate response, as well as fluxes of UV. Global average UV does not harm anything, but local increases are important.

9. Why do you average results over three years? What is special about that? I would be interested in the winter and summer seasons for each year, which is where the chemistry and dynamics responses determine the patterns. Three-year averages do not address the processes.

10. You use VEI as an index for the size of volcanic eruptions that affect climate, but that is wrong. Please see the explanation of why in Newhall et al. (2018), for example. Mount St. Helens, for example, was VEI 5, but had not sulfur and no impact on climate. I also recommend reading the original Newhall and Self VEI paper, which explains that it is an index of explosivity, and stratospheric inject is used as one criterion to assign VEI, but it should not be done in the opposite direction.

11. You use a mixture of different styles of reference, and they all need to be in the same style.

12. There were no supplemental figures in the manuscript you provided.

I provided 105 comments in the attached manuscript, all of which I recommend you address.

Please also note the supplement to this comment:
https://acp.copernicus.org/preprints/acp-2020-1110/acp-2020-1110-SC1-supplement.pdf
* * *
[Figure]

2020.

---

## Author Comment (AC1) · 1 Apr 2021

**Response to Reviews of "Co-emission of volcanic sulfur and halogens amplifies volcanic effective radiative forcing" by Staunton Sykes et al.**

We are very grateful to Daniele Visioni, an anonymous reviewer and Alan Robock for their comments and efforts which have helped us improve this manuscript. Following the structure recommended by ACP, we have responded to each reviewers' comments sequentially below with italicised and underlined text showing the reviewer's comments and plain text showing our response. Text which has been added to the manuscript is coloured red. Original manuscript text is in blue and any text which has been removed from the manuscript is blue and has been struck through. The locations of changes are stated. We hope these revisions address the comments of the reviewers.

**Review 1**

In this study the authors, using a CCM model, try to understand how co-emitted halogens may alter the climate impacts of stratospheric sulfur injection by volcanic eruptions. Their results show that the inclusion of halogens dramatically changes some of the simulated impacts, and that including such emissions is crucial to properly simulate large explosive volcanic eruptions. I found the paper to be excellent: the introduction does a great job at framing the problem and the methods are properly described. The results are also explained clearly, with pretty straightforward figures (which I suggest uploading in a higher quality format, they look a bit blurred when zoomed in). Overall, the manuscript is perfectly suitable to be published in ACP.

We thank Daniele Visioni for his comments and are pleased he assessed the work so highly. We acknowledge the suggestion to upload higher quality figures and have ensured this will be done before final submission.

**Specific Comments from Daniele Visioni:**

L 69: highlight with no s

Done.

**L 93: The correct name of the model is CESM1(WACCM) (Community Earth-System Model and then WACCM)**

The sentence beginning on line 93 has been amended accordingly to read:

Lurton et al. (2018) simulated the 2009 Sarychev Peak eruption (0.9 Tg of  $SO_2$ ) in CESM1(WACCM) (Community Earth-System Model, The Whole Atmosphere Community Climate Model (WACCM) and showed how inclusion of halogens...

L 116: from the way the phrase is written, it looks like the different halogen emissions are applied to the same amount of SO2, but in that case, the ratio would also have to be 100 times. But Ming et al. 2020 compare also a low and high SO2 injection (10 vs. 100 Tg) and in there different HCI concentrations. Just try to clarify this point.

The sentence beginning on line 116 has been amended accordingly to read:

They simulated 6 sets of experiments: a low SO2 (10 Tg) and high SO2 (100 Tg) eruption each paired with no HCI, low HCI (0.02 Tg) and high HCI (2 Tg), and found reported significant ozone depletion over both poles for at least four years in the high SO2 and high HCI experiment.that a volcanic halogen emission of 0.02 Tg (HCI:SO2 = 0.04) into a pre-industrial background state had little impact on column ozone but 2 Tg (HCI:SO2 = 0.4) showed significant and prolonged ozone depletion above both poles.

**Fig. 1: please specify what is the quiescent period against which the anomalies are calculated in the caption.**

We have amended the Figure 1 caption as follows:

Figure 1 - Global evolution of sulfur, halogens and OH for the SULF56, HAL56, SULF10 and HAL10 simulations, relative to the control climatology.

We have also amended the manuscript text in Section 2.2 Experimental Design to better explain which period the perturbations compared, as follows:

We utilise atmosphere-only, time-slice experiments whereby the initial sea surface temperature, sea ice fraction and forcing agents and depth, surface emissions and lower boundary conditions are prescribed using climatologies calculated using data from the fully coupled UKESM1.0 historical runs produced for CMIP6 (Eyring et al. 2016) and averaged over the years 1990 to 2000. By averaging over the decade the atmosphere-only simulations are forced with lower boundary conditions typical of the recent historical period but not a specific date within that decade, as desired. The fully coupled transient simulations had internally generated El Nino and La Nina cycles, however, averaging the SSTs over the 1990 to 2000 period resulted in a permanent neutral signal in the SST pattern, see figure S1. The 1990s, and thus these timeslices, were characterised by high background halogen levels due to anthropogenic emissions of CFCs throughout the preceding decade. The impacts of very short lived Bromine species are accounted for by adding a fixed contribution of 5 pptv into the CH3Br lower boundary condition.

A control simulation was run with a 15 year spin up followed by a further 20 years. A control simulation was initialised from the January 1995 initialisation file taken from the UKESM1.0 historical scenario which was run as part of CMIP6 (Eyring et al. 2016). The model was allowed to spin up for 15 years and the control was run for a further 20 years. The effect of the different explosive volcanic eruption scenarios (SULF10, SULF56, HAL10, HAL56) was investigated by running 6 10-year volcanic perturbation simulations for each scenario. The 6 simulations were initialised from 6 different years in the control run to represent the variability in QBO states. Changes are plotted as the difference between the average of the 6 ensembles and a climatology derived from the 20-year control run, cumulative forcings are calculated as the sum of the forcing over the full 10-year simulation duration.

Fig. 1f: I'm a bit confused as to why in the SULF simulations, there is a small increase in OH that I don't think is properly explained in the text. In the SO2 plume, we expect a large OH depletion. I assume that can be balanced out by the influx of water vapor in the stratosphere from the lower stratospheric heating and produce globally a slight increase. But I'd suggest checking (or consider the tropical changes in stratospheric OH, where I'm sure the change is negative – albeit less than in the HAL experiments).

In the initial submission, Figure 1f showed the global stratospheric mean OH percentage change. In sulfur-only simulations this was shown to increase due to the influx of water vapour into the stratosphere as a result of the tropical tropopause changes explored later in the paper. We have amended Figure 1f to show the tropical ( $20^{\circ}S-20^{\circ}N$ ) changes in stratospheric OH, which better shows the depletion of OH due to SO2 oxidation.

The accompanying caption has been amended accordingly to read:

(f) Tropical (20°N-20°S) sStratospheric OH change (%).

Line 255: it would be useful to show the changes in w\* (maybe in the supplementary next to Fig. S1) to show the difference in the transport induced by the stratospheric heating.

We have included an additional figure in the SI, showing a time series of the change in tropical (20°S-20°N)  $\overline{w}^*$  (residual mean vertical velocity) at 50 hPa (Figure S2).

**Figure S2**. Time series of 20°S - 20°N mean change in  $\overline{w}^*$  (residual mean vertical velocity) at 50 hPa.

**Fig. 3: I'd suggest switching panels a and b, as logically one might expect the lower injection scenario before.**

We agree with the reviewer that showing the smaller injection scenario first makes more logical sense. All plots and text have been amended accordingly.

Also, I find it interesting that the relationship doesn't hold as well for the lower injection case. I suspect this might be due to different QBO phases that affect the aerosols e-folding time (see Pitari et al., 2016), and that this effect is more evident for lower injection rates while for higher injection rates the increase heating rates modify the QBO too strongly independently on the phase it's in at the moment of injection (see for instance Aquila et al., 2014) resulting in similar lifetimes. The authors could just check if that's the case verifying the QBO phase, or just mention that's a possibility for the lower correlation in panel b (unless they have a better explanation).

SULF10, HAL10, SULF56, and HAL56 each have 6 ensemble simulations, initialised from the same 6 July initialisation files from the control simulation. Three of these initialisation files had a westerly QBO phase and the other three have an easterly QBO phase.

The correlation coefficient for regression lines is very high in both the large (r=0.88) and very-large eruption scenario (r=0.95). However, we acknowledge that the r value is slightly lower in the 10 Tg eruption scenarios, but suggest this is likely due to a smaller signal to noise ratio.

**Fig. 8: please specify if panels a-d are global changes**

The caption accompanying Figure 8 has been amended to read:

Figure 8 Evolution of global stratospheric mean water vapour (ppmv) in SULF56 and HAL56 (a), and SULF10 and HAL10 (b). Evolution of global stratospheric methane (ppmv) in SULF56 and HAL56 (c), and SULF10 and HAL10 (d).

**Review 2**

**General Comments:**

This study considers hypothetical VEI 6 and VEI 7 sized eruptions and, using a coupled chemistry-aerosol model driven by scenarios including or not including halogen injection, investigate how the co-emission of volcanic sulfur and halogens alters the evolution of the volcanic aerosol plume, stratospheric ozone chemistry, and the resulting radiative forcing and UV flux. The authors investigate how volcanic halogens may interact with the sulfur aerosol life cycle and interact to modulate volcanic forcing conversely to previously reported work. I found the link between chemical and microphysical processes of particular interest. The authors reveal in their model experiments the primary importance of halogens in major volcanic emissions in the sulfur cycle in the stratosphere, a process already suspected for eruptions of much more minor amplitude. Impacts of halogen emissions on dynamics of the aerosols and the subsequent effect on aerosol microphysics are also considered here when critically missing in previous reported studies. Effects on some key stratospheric compounds like ozone, water vapour and methane are also analysed. To me, this work points at the critical need to maintain space-borne observations of stratospheric compounds which will be particularly valuable to quantify halogen injected by volcanoes and needed for model initialization. The authors finally mention open questions to be addressed in future studies reflecting the great interest to consider these events and their associated various injections (sulfur, halogens, ash, water vapour) in the future climate. This study is original and comprehensive judging by the various topics and impacts covered (microphysics. dynamics, chemistry and radiative forcing). I found the manuscript clear, well-written and nicely going straight to the obtained results. I estimate that this work deserves to be published in ACP after the following minor comments have been addressed.

We are glad that the reviewer found the manuscript clear and concise, while remaining original and comprehensive. We are thankful for their useful and constructive comments which have helped to improve this manuscript.

**Specific Comments from Reviewer 2:**

Introduction: I am not a specialist of petrological processes but could you indicate the degree of uncertainty when petrological budgets are used to derive stratospheric inputs of halogens? What are the assumptions behind the halogen injection efficiency (I would suggest to briefly recall the definition of the stratospheric halogen injection efficiency).

The stratospheric halogen injection efficiency is the fraction of the halogens degassed from the magma at the vent that are transported into the stratosphere. Textor et al. (2003b) used plume rise models to suggest that the halogen injection efficiency of explosive volcanic eruptions is 10 to 20%. We calculated halogen injection efficiencies for past eruptions using petrological data as an estimate of the amount of halogens degassed at the vent and then used ice core and satellite data to estimate the amount that made it into the stratosphere.

P5 lines 158-168: How the overall chemical species initialized? Is it based on climatological 3D fields or only surface emissions provided by CMIP6? Especially for bromine compounds, how the Bry budget initialized? Are very-short-lived species accounted for? The resulting inorganic bromine budget in the stratosphere and more generally the inorganic halogen content, computed in chemistry models is of particular importance regarding ozone chemical cycles and would have significant impact in your scenario with no volcanic emissions of halogens. Please provide a bit more information.

All chemical species in our model set up are initialised from the 3D January 1995 initialisation file from the UKESM1.0 historical scenario which was run as part of CMIP6. During the simulations, surface emissions of BC,  $C_2H_8$ ,  $C_5H_8$ , CO, DMS, HCHO, Me2CHO, Monoterp, NH3, NO, NVOC, OC and anthropogenic SO2 are included following the CMIP6 UKESM1.0 historical scenario and the concentrations of CO2, CH4, H2, N2O, LLCI and LLBr are specified as surface concentrations following the CMIP6 UKESM1.0 historical scenario (Eyring et al., 2016). The impacts of VSLBr are accounted for by adding a fixed contribution of 5pptv into the CH3Br surface concentration.

**Where:**

Long lived Chlorine (LLCI) = CH3CI, CH3CCI3, CCI4, CFC-11, CFC-12, CFC-113, CFC-114, CFC-115, HCFC-141b, HCFC-142b, HCFC-22

Long lived Bromine (LLBr) = CH3Br, H-1301, H-1211, H-1202, H-2402 Very Short Lived Bromine (VSLBr) = CHBr3, CH2Br2, CH2BrCl, CHBrCl2, CHBr2Cl, CH2IBr

Section 2.2 has been amended, as follows:

We utilise atmosphere-only, time-slice experiments whereby the initial sea surface temperature, sea ice fraction <del>and forcing agents</del> and depth, surface emissions and lower boundary conditions are prescribed using climatologies calculated using data from the fully coupled UKESM1.0 historical runs produced for CMIP6 (Eyring et al. 2016) and averaged

over the years 1990 to 2000. By averaging over this timeframe the atmosphere-only simulations are forced with boundary conditions typical of the recent historical period but not a specific date within that decade. The fully coupled transient simulations had internally generated El Nino and La Nina cycles, however, averaging the SSTs over the 1990 to 2000 period resulted in a permanent neutral signal in the SST pattern, see figure S1. The 1990s, and thus these timeslices, were characterised by high background halogen levels due to anthropogenic emissions of CFCs throughout the preceding decade. The impacts of very short lived Bromine species are accounted for by adding a fixed contribution of 5 pptv into the CH3Br surface concentration.

A control simulation was run with a 15 year spin up followed by a further 20 years. A 20-year control simulation was run after a 15 year spin up, initialised from the January 1995 initialisation file taken from the UKESM1.0 historical scenario which was run as part of CMIP6 (Eyring et al. 2016). The effect of explosive volcanic eruptions was investigated by running a series of 10 year volcanic perturbation simulations spun off from 6 different years in the control run to represent the variability in QBO states. Changes are plotted as the difference between the average of the 6 ensembles and a climatology derived from the 20 year control run, cumulative forcings are calculated as the sum of the forcing over the full 10 year simulation duration.

**P5 line 170 onwards: You do not differentiate between SW and LW radiation conversely to the work of Schmidt et al. (2018). What justifies this choice?**

We have included an additional figure into the SI showing a breakdown of the TOA  $\text{ERF}_{ari}$ ,  $\text{ERF}_{cc}$ , and total volcanic ERF anomalies into the contributing shortwave and longwave changes (Figure S4), and amended the manuscript text as follows:

The radiative impact of sulfate aerosols depends on the particle size, amongst other things (Timmreck et al., 2010, Pinto et al. 1989). Using Mie scattering theory, (Lacis, (2015) found that the scattering cross section per unit mass at 550 nm is largest for sulfate aerosol with effective radius of ~0.20  $\mu$ m. The smaller Reff in HAL10 and HAL56, compared to SULF10 and SULF56, is closer to 0.20  $\mu$ m and results in more efficient scattering of SW of radiation per unit mass (Timmreck et al., 2010). Therefore, we simulate 11% and 22% higher peak global-mean stratospheric aerosol optical depth (SAOD) anomalies at 550 nm in HAL10 and HAL56 than their equivalent SULF simulations (Figure 4), despite having a 14% and 9% lower smaller peak aerosol burden. Correspondingly, we simulate an 8% and 6% increase in the peak global-mean ERFari in HAL10 and HAL56 compared to SULF10 and SULF56 (Figure 4), driven by a 14% and 11% increase in peak global-mean SW forcing (Figure S4). The SAOD and ERFari anomalies are a balance between the offsetting effects of smaller aerosol and shorter lifetime which result in a net-zero impact on cumulative ERFari despite a significant increase in the peak global-mean ERFari (Figure S2a,b).